# Advances in Surface Water and Ocean Topography for Fine-Scale Eddy Identification from Altimeter Sea Surface Height Merging Maps in the South China Sea

Xiaoya Zhang[1, 2], Lei Liu[1, 2], Jianfang Fei[1], Zhijin Li[3], Zexun Wei[4], Zhiwei Zhang[5], Xingliang Jiang[3], Zexin Dong[1], Feng Xu[1]

[1]College of Meteorology and Oceanology, National University of Defense Technology, Changsha 410073, China

[2]Key Laboratory of High Impact weather (special), China Meteorological Administration, Changsha 410073, China

[3]Department of Atmosphere and Ocean Sciences, Institute of Atmospheric Sciences, Fudan University, Shanghai 200433, China

[4]First Institute of Oceanography, and Key Laboratory of Marine Science and Numerical Modeling, Ministry of Natural Resources, Qingdao 266061, China

[5]Frontier Science Center for Deep Ocean Multispheres and Earth System (FDOMES) and Physical Oceanography Laboratory/Key Laboratory of Ocean Observation and Information of Hainan Province, Sanya Oceanographic Institution, Ocean University of China, Qingdao/Sanya, China

*Correspondence to*: Lei Liu (liulei17anj@nudt.edu.cn)

**Abstract.** The recently launched Surface Water and Ocean Topography (SWOT) satellite mission has reduced the noise levels and increased resolution, thereby improving the ability to detect previously unobserved fine-scale signals. We employed a method to utilize the unique and advanced capabilities of SWOT to validate the accuracy of identified eddies in merged maps of a widely used Archiving, Validation, and Interpretation of Satellite Oceanographic (AVISO) data product and a newly implemented two-dimensional variational method (2DVAR), which uses a $1/12°$ grid and reduces decorrelation of spatial length scales. SWOT data are more likely to provide detailed comparisons of eddy boundaries for fine- to meso-scale structures compared with conventional in-situ data (e.g., drifting buoys). The validation results demonstrated that compared with AVISO, the 2DVAR method exhibited greater consistency with the SWOT observations, especially at small scales, confirming the accuracy and capability of the 2DVAR method in the reconstruction and resolution of fine-scale oceanic dynamical structures.

# 1 Introduction

The ocean has diverse spatial length scales of dynamical processes, from mesoscale signals at approximately 100–1000 km to sub-mesoscale processes below 100 km. Fine-scale ocean processes are characterized by spatial variability of 1 to 100 kilometres and temporal variability of days to months (Lévy et al., 2024). These processes are primarily revealed through the absolute dynamic topography (ADT), which is an estimate of sea surface height (SSH) above the geoid. ADT also plays a substantial role in the thermohaline circulation, atmosphere–ocean interactions, physical–biological–biochemical interactions, and the numerical modeling of coupled atmospheric-oceanic systems (Chelton et al., 2007; Ma et al., 2016; Mahadevan, 2016; Wunsch and Heimbach, 2013).

Global satellite altimeters offer systematic ADT measurements and mapping of ocean topography, currently providing the most effective data for detecting and tracking large and mesoscale ocean dynamic signals (Chelton et al., 2007; Mason et al., 2014; Zhang et al., 2023). Due to differences in orbit cycles and swaths from different satellites, the observing data exhibit misalignment in both space and time. Consequently, ensemble Kalman filtering or data assimilation techniques based on optimal estimation methods are employed to merge data from different satellites, yielding a spatiotemporally continuous ADT map (Cohn, 1997; Le Traon et al., 1998; Taburet et al., 2019). The main techniques of data assimilation typically include the homogenization and cross-calibration of multi-source altimetry data, continuous calibration of reference orbits, cross-calibration between altimeters, long-wavelength error correction, and error budget modeling. Finally, optimal interpolation is used for gridding to generate daily gridded products and derived products (Pujol et al., 2016). Diverse merging methods result in disparate capacities for capturing oceanic dynamic signals which can be assessed by the metrics such as effective resolution and eddy kinetic energy (Ballarotta et al., 2019, 2020; Pascual et al., 2007; Taburet et al., 2019; Wang et al., 2021). Those assessment methods that rely on conventional measurement data are inherently limited by linear and long temporal sampling, low resolution, and other observational uncertainties, making them unsuitable for assessing merged maps in regions characterized by intricate multiscale oceanic dynamic signals.

The Surface Water and Ocean Topography (SWOT) satellite, launched in December 2022, comprises a new generation of Ka-band radar interferometers (KaRIn), which reduces instrument noise by two orders of magnitude compared to that of the conventional satellites (Abdalla et al., 2021; Fu et al., 2024). The KaRIn technique allows the mapping of two-dimensional ADT with a 120-km wide swath, which is over five times the width of a conventional nadir, and offers an unprecedented 15-km spatial resolution for an altimeter satellite (Dufau et al., 2016; Morrow et al., 2019; Wang and Fu, 2019). Globally, SWOT data has undergone in-situ observational calibration and validation and data assimilation application studies before its formal use for global mapping, confirming the capability of detecting previously unobserved fine-scale signals, reinforcing the capabilities of ocean monitoring, signifying a pivotal advancement in enhancing spatial resolution (Martin et al., 2024; Ubelmann et al., 2024; Verger-Miralles et al., 2024; Zhang et al., 2024). However, the intrinsic challenge posed by the discrepancy between low temporal and high spatial resolution requires further interpretation before direct utilization as inputs for ADT merged maps in future research endeavours.

This study aimed to validate the accuracy and reliability of different merging methods, specifically 2DVAR and AVISO, in reconstructing oceanic dynamic signals, with a particular focus on fine-scale eddies. It introduces a novel application scenario and methodology for utilizing state-of-the-art international sea surface observation data. Furthermore, it offers a new framework for assessing the quality of merged maps, which can provide valuable insights and guidance for the development of future merging techniques.

## 2 Materials and Methods

## 2.1 ADT Merging Maps

The SWOT mission consists of two phases: the science phase, which conducted 21-day repeat sampling from September 7[th] 2023 to November 21[st] 2023, and the calibration and validation phase (CALVAL), which performed 1-day rapid sampling from April 1[st] 2023 to July 31[st] 2023 (AVISO/DUACS, 2024). The CALVAL phase data were used exclusively in the second part of Section 3.2 to analyse the temporal evolution of eddies. In contrast, the science phase data were the primary datasets for examining the spatial dynamic structures and for performing statistical analyses of eddy characteristics. Although the CALVAL phase sampled a limited sea surface area due to its fixed rapid-sampling orbit, this orbit covered part of the South China Sea (SCS) and facilitated the capture of time-evolving fine-scale eddy structures in the SCS. The nadir observation points, located between two slices of KaRIn observations (Fig. 1(a)), were excluded in both phases due to their high error rates and our focus on the advanced KaRIn technology. To ensure consistency in data resolution and to focus the current study on the fine-scale to mesoscale, we employed a regional averaging method to reduce the resolution of the SWOT data from the original 2-km sampling interval to 1/12°. Owing to the inclination angle between the SWOT satellite orbital plane and the equatorial plane, each downsampled square region is covered by observations. Consequently, interpolation across swath gaps is unnecessary, thereby avoiding the substantial errors that associated with interpolating these gaps.

We employed two types of ADT merged data to generate corresponding eddy identification ensembles. The first ADT product (Fig. 1b) was produced using a $\pm 11$ days time window Near Real-Time (NRT) Two-Dimensional Variation (2DVAR) method with a 1/12° grid resolution. It employs optimized background and observation errors to decorrelate the length scales in the merging process, with an improved method for calculating the background error covariance matrix (Liu et al., 2020). The second ADT merged map (Fig. 1c) was a product released by AVISO, which uses the optimal interpolation method (Pujol et al., 2016) with a global 1/4° grid resolution. During the science phase of the SWOT mission, the AVISO merged map Delayed Time (DT) products utilized SWOT nadir data as an input source (Copernicus Marine Service repository, 2023b). To maintain the independence of the datasets, we employed NRT products, which do not include SWOT nadir data (Copernicus Marine Service repository, 2023a). Similarly, during the CALVAL phase of the SWOT mission, we used an older version of the DT products which do not include SWOT data as an input source. The DT products are reanalysis datasets that incorporate the highest quality altimeter measurements and geophysical corrections to minimize the

risk of mass loss or false signals over time. The NRT data provide ready-to-use, real-time published altimeter data from all available missions. In the data processing, the DT products are computed optimally using a centered computation time window of $\pm 6$ weeks around the date of the map to be computed. In the NRT processing, future data are not available; therefore, the computation time window covers the period from 7 weeks prior to the computation date. Both AVISO and 2DVAR methods were based on the principle of optimal estimation and were calculated using all available on-orbit altimeter mission data, including Jason-3, Sentinel-3A, HY-2B, Saral/AltiKa, Cryosat-2, Sentinel-3B, and Sentinel-6A. Notably, the 2DVAR method successfully reduced the effective resolution to approximately half that of AVISO, enhancing the resolution of signals from fine-scale to mesoscale eddies, particularly in areas with rich eddy kinetic energy, such as the SCS and the Northwest Pacific Ocean (Jiang et al., 2022; Liu et al., 2023).

The SCS is a significant dynamic marginal sea in the northwestern Pacific, featuring complex bathymetry, a large area, and multiple straits that facilitate water exchange with the Pacific and Indian Oceans(Chen et al., 2023). It serves as an exemplary model of an open ocean with well-defined continental shelves, shelf breaks, and a central deep basin. In the SCS, the first obliquely pressured Rossby deformation radius was less than 20 km in winter (Cai et al., 2008), suggesting a rich environment for fine-scale oceanic dynamical processes. Additionally, the SCS receives energy transport from the sub-mesoscale energy reservoir of the Kuroshio Current via the western boundary currents, resulting in a dense concentration of mesoscale and fine-scale processes on the 10-km scale (Lin et al., 2020; Ni et al., 2021; Zu et al., 2019). Thus, this study of the SCS holds substantial significance and reference value for understanding complex dynamic marginal seas and the broader northwestern Pacific region.

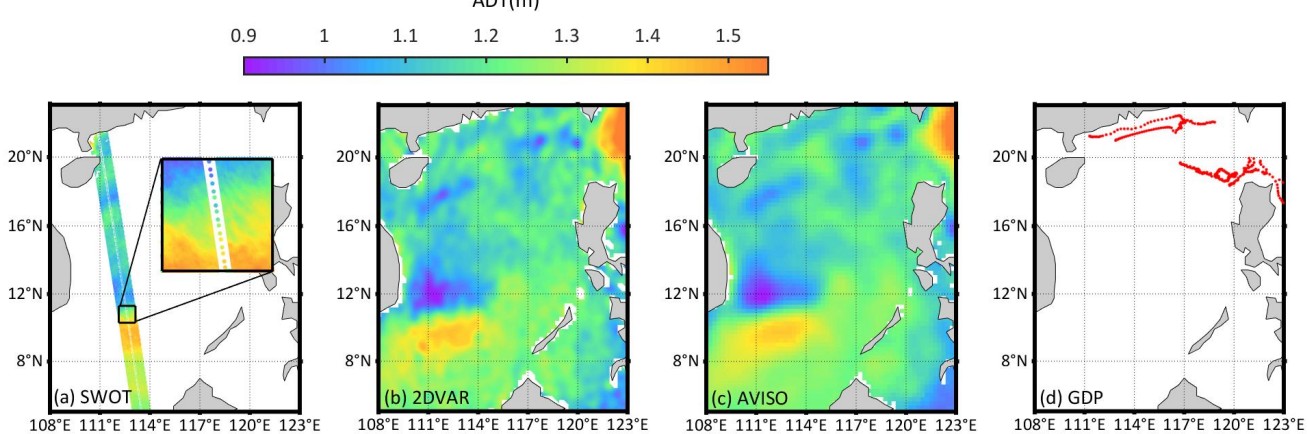

Figure 1. Four datasets of absolute dynamic topography (ADT) in the South China Sea. (a) SWOT, (b) 2DVAR, (c) AVISO, and (d) GDP. The ADT data in (a), (b), and (c) were obtained on September 12, 2023, and (d) covers the entire period of the Science phase.

## 2.2 Eddy Identification

Currently, the main methods for eddy identification based on satellite altimeters include the Okubo-Weiss (OW) parameter from the velocity field method, the curvature center method, the surrounding angle method, the local extreme of sea surface

topography method, the local and normalized angular momentum method, and the Lagrangian coherent structure (LCS) method (Chelton et al., 2011; Laxenaire et al., 2018; Mcwilliams, 1990; Mkhinini et al., 2014; Okubo, 1970; Sadarjoen and and, 2000; Weiss, 1991). Among these, the sea surface topography method provides the clearest and most cohesive identification of eddies, regardless of their size or boundary (Chen et al., 2021). Therefore, this research employed a sea surface topography method based on contour analysis for eddy identification in 2DVAR and AVISO ADT merged maps, as well as in SWOT maps during both phases of the SWOT mission (Chelton et al., 2011). The eddy tracking was adopted only during science phase because of the fixed observation area of CALVAL phase. It is worth noting that, due to SWOT observation data limitations, we are currently unable to identify eddies located at the edges of the swath or outside the swath. To avoid the influence of grid resolution, different merged maps were interpolated to a high-resolution grid with the same resolution (1/12°). The outermost circle of the closed contours with 1mm step in ADT difference containing the unique centre was recognized as the 'quasi-eddy edge', and only a minimum of three points were retained. Each quasi-eddy edge was then contracted inward until it corresponded to a single centre. Lastly, the geometric centre of the innermost circle of the closed contours was identified as the eddy centre. This process allowed the determination of the eddy boundary, type, radius, and amplitude. All possible eddies with a difference in ADT between the eddy center and the boundary contour line of less than ±2 cm were excluded from further analysis.

## 2.3 Eddy Validation

The accuracy and reliability of the identified eddies were validated and assessed using two observational ADT datasets as true values. First, the Level-3 LR ADT v1.0.2 datasets from the SWOT product were used for eddy validation, and a visualization method was employed to examine the clarity and intuitiveness of the eddy boundaries. The Level-3 data are more suitable for capturing fine-scale to mesoscale structures compared to the Level-4 products, which rely on merging methods and data from other satellites (Ballarotta et al., 2023). The 2-km sampling interval allowed the SWOT data to capture finer-scale signals that are not the primary focus of the current research. Therefore, spatial average filtering with a 1/12° grid was conducted to filter out those finer-scale signals (Fig. 1a), highlighting the fine-scale to mesoscale eddies in this study.

Before the validation process, all identified eddies were preliminarily evaluated to ensure they could be matched with a specific SWOT eddy. Once a successful match is established, the eddy reconstruction of the ADT map will be considered correct and will be retained along with its corresponding eddy captured from the SWOT map. Only eddies that met the following three criteria were considered to match the SWOT eddy, which was assumed to be actual:

1) The rotation direction or type of the merged-map eddy and the SWOT eddy are the same
2) The distance between the centre of the merged-map eddy and the SWOT eddy is less than 50 km
3) The difference in radius between the merged-map eddy and the SWOT eddy is less than 120 km

The criteria were set according to the KaRIn swath, which is approximately 120 km for the full swath and approximately 50 km for the half swath. After filtering out eddies that cannot be matched with SWOT, a systematic standardization process

was adopted to make merged-map eddies comparable with the SWOT eddies in terms of spatial scales and relative location (Chen and Yu, 2024). The operation aims to standardize SWOT eddies of diverse sizes and proportionally scale the eddies in the 2DVAR and AVISO merged maps relative to those in SWOT, thereby retaining the relative differences between the merged maps and SWOT. This normalization and proportional scaling allow the statistical synthesis of differences among thousands of eddies into a single standard grid composite map, effectively characterizing the radius discrepancies between the merged maps and SWOT eddies. Firstly, the scale normalization factor $\alpha$ was calculated for each SWOT eddy to adjust it to a fixed size based on the eddy radius:

$$\alpha_i = \frac{\frac{1}{N}\sum_{i=1}^{n} r_i}{r_i} \quad (1)$$

where $i$ is the index of the $i$th eddy in the SWOT ensemble, $N$ denotes the total number of the eddies, and $r$ represents the radius of the eddy. To eliminate the absolute positional differences, we established a local coordinate system with the centre of normalized SWOT eddy as the origin. The geographic coordinates of the $i$-th SWOT eddy centre ($X_{i,S}$, $Y_{i,S}$) were subtracted from those of the corresponding merged-map eddy centres ($X_{i,c}$, $Y_{i,c}$) and boundary points ($X_{i,bm}$, $Y_{i,bm}$) :

$$\begin{cases} \left(X'_{i,c}, Y'_{i,c}\right) = \left(X_{i,c} - X_{i,S}, Y_{i,c} - Y_{i,S}\right) \\ \left(X'_{i,bm}, Y'_{i,bm}\right) = \left(X_{i,bm} - X_{i,S}, Y_{i,bm} - Y_{i,S}\right) \end{cases} \quad (2)$$

the new centre and boundary points of the $i$th merged-map eddy in the local coordinate system are ($X'_{i,c}$, $Y'_{i,c}$) and ($X'_{i,bm}$, $Y'_{i,bm}$), where $bm$ denotes the index of the boundary point.

Based on the coordinate system transformation, the position information of the merged-map eddies under the new coordinate system was scaled using the SWOT eddy normalization factor $\alpha$. Considering that eddy scaling is related to the radial distance, this step applied a polar coordinate transformation to extract the radial distances $r_{i,c}$, $r_{i,bm}$ and angles $\theta_{i,c}$, $\theta_{i,bm}$ of the points:

$$\begin{cases} r_{i,c} = \sqrt{\left(X'_{i,c}\right)^2 + \left(Y'_{i,c}\right)^2} \\ \theta_{i,c} = \arctan\left(\frac{Y'_{i,c}}{X'_{i,c}}\right) \\ r_{i,bm} = \sqrt{\left(X'_{i,bm}\right)^2 + \left(Y'_{i,bm}\right)^2} \\ \theta_{i,bm} = \arctan\left(\frac{Y'_{i,bm}}{X'_{i,bm}}\right) \end{cases} \quad (3)$$

By multiplying the coordinates with the normalization factor $\alpha$, we implemented the scaling. The scaled coordinates were then transformed back to obtain geometrically similar coordinates ($X''_{i,c}$, $Y''_{i,c}$) and ($X''_{i,bm}$, $Y''_{i,bm}$).

$$\begin{cases} X''_{i,c} = r_{i,c} \cdot \alpha_i \cdot \cos(\theta_{i,c}) \\ Y''_{i,c} = r_{i,c} \cdot \alpha_i \cdot \sin(\theta_{i,c}) \\ X''_{i,bm} = r_{i,bm} \cdot \alpha_i \cdot \cos(\theta_{i,c}) \\ Y''_{i,bm} = r_{i,bm} \cdot \alpha_i \cdot \sin(\theta_{i,bm}) \end{cases} \quad (4)$$

To compare with SWOT, this section also describes how the validation of eddies using traditional in-situ drifter observations.

The Global Drifter Program (GDP) data provide the positions of a 15-m depth drogue drifter at a 1-hour frequency and have been frequently employed in the validation and assessment of surface dynamic signals (Zhang and Qiu, 2018). These data enable studies of fine-scale to mesoscale dynamical structures, comparative evaluation of global ocean numerical models, and provide input data for forecasting (Lumpkin and Elipot, 2010; Yu et al., 2019). Due to its Lagrangian nature, the drifter is more likely to respond to low pressure and high velocity in the central region of the eddy, and then become entrained

within the eddy as it rotates toward the centre (Ohlmann et al., 2017). Thus, the trajectory of the drifter allows for the observation and capture of fine-scale and mesoscale eddies in this research. The region of interest in this study is situated within the convergence zone of the subtropical circulation, it contains several drifter sampling areas and involves a long drift time. However, the GDP datasets are only available during the scientific phase of the SWOT mission (Fig. 1d).

## 3 Results

### 3.1 Accuracy and Reliability of Identified Eddies

In this section, composite maps of normalized eddy ensembles for 2DVAR and AVISO ADT merged maps will be presented first, followed by a summary of the errors and characteristics of the identified eddies.

The coloured areas in Fig. 2 represent the distributions of the normalized radii of eddies identified by 2DVAR and AVISO, which were successfully matched with SWOT eddies. All identified eddies were categorized into three groups based on the

radii of SWOT eddies: those with radii below 10 km, between 10 km and 20 km, and exceeding 20 km were classified as fine-scale (Fig. 2a), submesoscale (Fig. 2b), and mesoscale (Fig. 2c) eddies, respectively. The dashed circle lines in each component represent the normalized SWOT eddies, and the size of the coloured area outside these dashed circles illustrates the discrepancy between the merged-map eddies and SWOT eddies. A grid space extent equal to twice the normalized SWOT eddy radius was chosen.

Despite geographical and radius distribution discrepancies, the merged map shows a certain degree of accuracy and similarity with SWOT in Fig. 2. As the eddy scale increases from Figs. 2a to 2c, the maxima on the eddy composite maps are situated closer to the origin, and the error proportions beyond the dashed circles decrease. This suggests that the merged map is more accurate at reconstructing mesoscale eddies compared to fine-scale eddies.

For reconstructing the same SWOT eddy categories, the two merged maps exhibit different performance. Since the same

colorbar is applied to the eddy ensembles of the two merged maps for the same scale range, the distribution and

concentration of eddies can be judged by the intensity of the colors. It is evident that the area of the colored region outside the normalized SWOT eddies (marked by black dashed circles) in the 2DVAR merged map is smaller especially in Fig. 2b and 2c. Meanwhile, despite the number of eddies identified by 2DVAR being 2 to 3 times that of AVISO, the color outside the normalized eddy circles remains a light shade of purple (means no more than 10 eddies). These results suggest that the concentration of 2DVAR eddies within the normalized SWOT eddies is higher, and the eddy boundaries maintain a higher degree of consistency. Additionally, 2DVAR results in more matches with SWOT across all categories, particularly for SWOT eddies with scales less than 10 km (Fig. 2a), where the number of matches is three times greater than those achieved by AVISO.

For all matched eddies, the average radius of the matched SWOT, 2DVAR, and AVISO eddies is approximately 20 km, 40 km, and 65 km, respectively, indicating a significant difference between the merged maps. The size range for all matched 2DVAR eddies, from 15 to 100 km, is closer to that of SWOT (ranging roughly from 8 to 54 km) than the broader span of 15 to 134 km for AVISO. Compared to the normalized eddies from AVISO, that of 2DVAR are more concentrated, and their boundaries are more precise relative to the dashed circle representing the actual SWOT eddies. This indicates that 2DVAR performs better than AVISO in reconstructing smaller eddies. We also calculated the eddy identification ratio based on the eddy quantity in the merged map compared to that in the SWOT data. The results demonstrate that as the SWOT observation radius increases, the eddy identification ratio of the 2DVAR method rises from 25% to 40%, while the identification ratio of AVISO remains relatively stable at around 11%. This leads to a significant increase in the gap between the two methods, from 2.5 times to 4 times. This contrast highlights the superior performance of the 2DVAR method in detecting eddies using SWOT data, especially in capturing fine-scale to mesoscale features that AVISO may miss.

Table 1. Root mean square error (RMSE) and Extremes: Eddy Radius, Amplitude, Centre Location, and identified ratio (2DVAR, AVISO compare to SWOT).

| | SWOT Categories | 2DVAR | AVISO |
|---|---|---|---|
| RMSE of Radius [km] | | 25.13 | 47.33 |
| RMSE of Amplitude [cm] | | 2.89 | 3.53 |
| RMSE of Centre Location [km] | | 25.72 | 26.90 |
| Max./Min. radius of corresponding SWOT eddy [km] | | 55.71/5.38 | 56.24/7.96 |
| Max./Min. radius [km] | fine-scale | 48.09/15.13 | 69.75/15.42 |
| | submesoscale | 101.23/15.63 | 128.57/16.29 |
| | mesoscale | 89.06/18.37 | 134.52/23.70 |
| Eddy identification ratio | fine-scale | 25.0% | 10.0% |
| | submesoscale | 36.7% | 12.2% |
| | mesoscale | 39.1% | 11.2% |

Note: In row four and five, the "Max./Min. radius" specifies the range of merged-map eddy radii for each SWOT category corresponding to Figure 2, with the leading number being the maximum and the trailing number being the minimum radius.

The discrepancies in eddy radius between the merged maps and SWOT maps should not be ignored. To provide a more detailed assessment, the root mean square error (RMSE) for the eddy radius, amplitude, and centre location was also summarized in Tab. 1. Among these statistics, the RMSE in eddy radius, especially for AVISO, is comparable to the SWOT radius itself, illustrating that the spatial scale is a significant issue in the ADT merged map. The 2DVAR merging method has reduced the radius error by approximately 20 km compared to AVISO, mostly attributed to its accurate capturing of

mesoscale eddies. The amplitude error is about 3 cm for both merged maps, which is half the maximum amplitude of the SWOT eddies (as shown in Section 3.2). The center location error was calculated using the physical location, which is caused by the positional deviation of local maxima or minima in ADT signals. However, the errors in amplitude and centre position do not show a notable improvement in 2DVAR compared to AVISO.

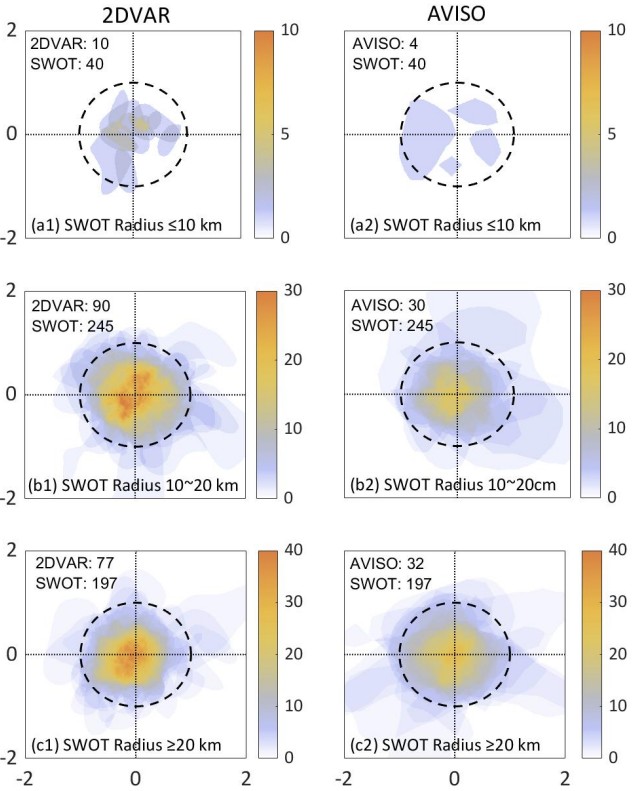

**Figure 2. Composite maps of normalized eddy identified from 2DVAR (left column) and AVISO (right column) merged maps, the color on the grid points shows the density of covered eddies, with the higher the density of the grid points, the darker the orange. The dashed circles mark the normalized SWOT eddies with a radius of less than 10 km (a), 10 to 20 km (b), and more than 20 km (c). The total number of eddies detected by each map is on the left up corner.**

## 3.2 Eddy Boundaries Verification in Space and Time

This section presents a detailed visual display of merged-map eddies compared with SWOT eddies and the eddy boundaries with drifter trajectories to further validate the reliability and accuracy of the identified eddies.

Figure 3 provides a detailed comparison of eddies from two merged maps using science-phase SWOT mission data in the central area of the SCS. The coloured points in the figure shows the Level-3 ADT observations directly from the KaRIn measurement on SWOT satellite. A high degree of agreement was found between the eddies identified using 2DVAR and

245 SWOT at scales ranging from 50 to 200 km. The 2DVAR product demonstrates strong agreement with SWOT-derived anticyclonic eddies at 116°E, 21°N in Fig. 3(a1), 112°E, 17.5°N and 112°E, 16°N in Fig. 3(b1), and 113°E, 18°N in Fig. 3(c1), as well as with the cyclonic eddy at 111.5°E, 16.5°N in Fig. 3(b1). In contrast, the AVISO product fails to accurately match the cyclonic eddy at 111.5°E, 16.5°N with the SWOT observations in Fig. 3(b2). Additionally, the AVISO product captures an eddy at 116°E, 21°N in Fig. 3(a2) that only partially overlaps with the corresponding SWOT eddy, with minimal

spatial correspondence. For other eddies where both merged products exhibit agreement, the radii of the AVISO eddies are notably larger than those identified by 2DVAR. Although the AVISO product exhibits lower consistency with SWOT observations compared to 2DVAR, it still captures a considerable number of eddies that align with SWOT, thereby maintaining its fundamental utility as a merged product for eddy identification. However, both products fail to detect certain small eddies in SWOT observations, such as the cyclonic eddy at 112.5°E, 17°N in Fig. 3(c).

It is important to emphasize that the examples presented in this section are representative of the eddy boundary reconstruction capabilities of both AVISO and 2DVAR. While these examples show that AVISO performs slightly worse than 2DVAR in terms of eddy radius and positional accuracy, they nonetheless represent some of the best-case scenarios for the AVISO product. This conclusion is not influenced by selective bias in the examples chosen but rather reflects the inherent performance differences between the two merged products, which is consistent with the statistical findings

presented in section 3.1.

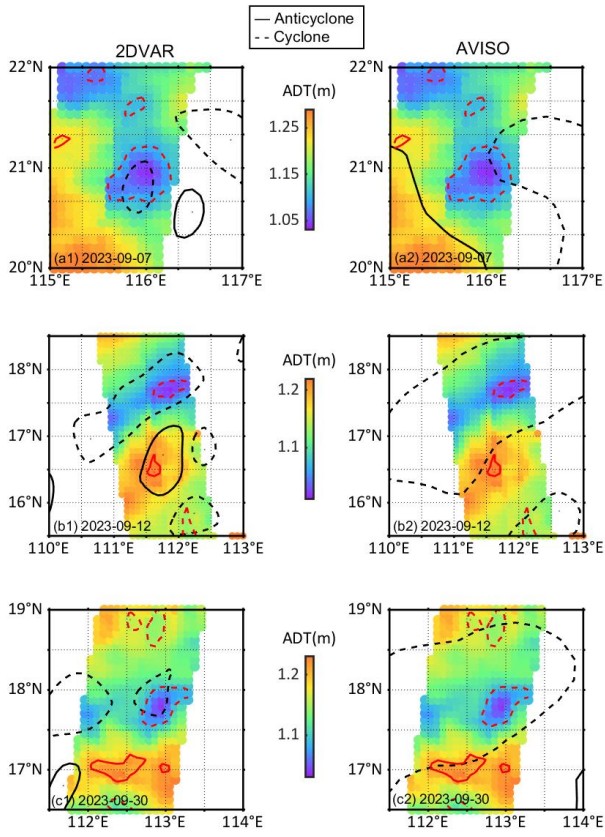

**Figure 3. The ADT observation data of SWOT with the eddies detected by SWOT (in red), 2DVAR (in black, a1-c1) and AVISO (in black, a2-c2). The solid line represents the anticyclonic eddies and dashed line represents the cyclonic eddies.**

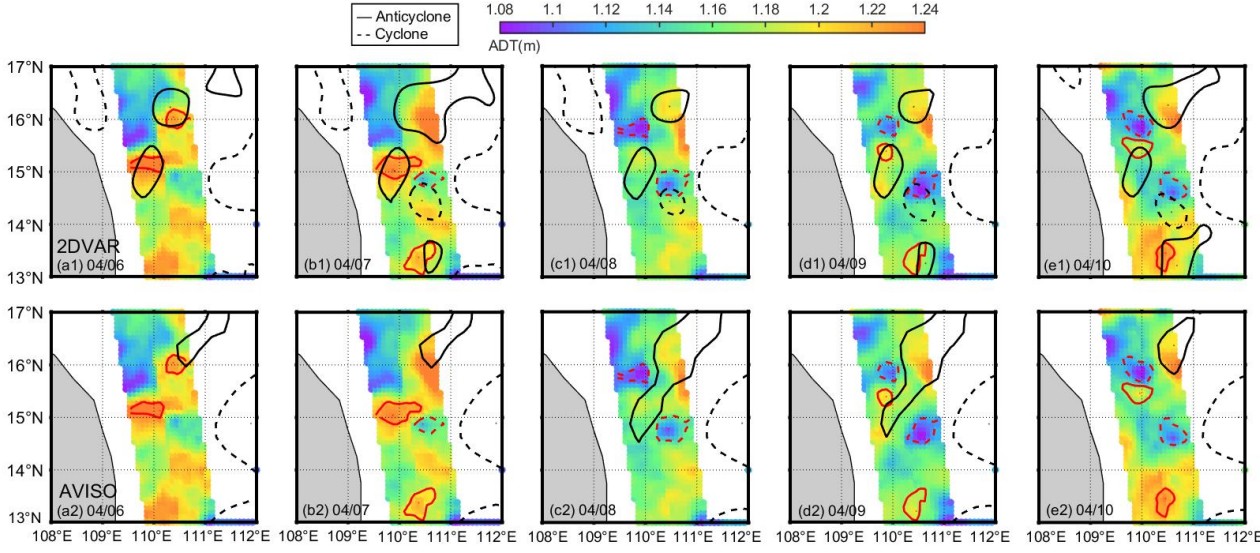

**Figure 4. Observation data of SWOT (in red) with the 2DVAR (in black, a1-e1) and AVISO (in black, a2-e2) merged maps from**

**2023/04/06 to 04/10. The solid line (dashed) represents the anticyclonic (cyclonic) eddy, and the colour-filled plot contains the KaRIn data from SWOT.**

The CALVAL phase is utilized to capture the evolution of small-scale structures over time, distinct from the science phase. A duration of five days was selected for this section, corresponding to the period of fine-scale structure evolution.

In the 2DVAR maps, two closely spaced mesoscale anticyclones were identified at 110°E, 15°N, and 110.5°E, 16.2°N in Figs. 4(a1)–(e1), along with an anticyclone at 110.5°E, 13.2°N in Figs. 4(b1, d1 and e1), and a smaller-scale cyclone at 110.5°E, 14.8°N in Figs. 4(b1)–(e1). These eddies derived from 2DVAR exhibit discrepancies when compared to the SWOT-derived eddies, particularly in the case of the eddy at 110.5°E, 16.2°N whose radius varies daily in the 2DVAR results and represented by a colored circle on the map rather than being identified as a distinct eddy in the SWOT data.

Similar to as Fig. 3, the 2DVAR method outperforms AVISO, which erroneously merges the two closely spaced anticyclones into a single larger eddy and fails to capture the eddies at 110.5°E, 14.8°N and 110.5°E, 13.2°N. Notably, by accurately matching the emergence and dissipation of SWOT eddies at 110.5°E, 14.8°N and 110.5°E, 13.2°N, the 2DVAR method demonstrates its capability to reconstruct eddies that evolve over time, despite some relative positional deviations from the actual eddy location.

Additionally, in the underlying SWOT data, although the algorithm fails to identify an eddy that is only partially within the swath, the color map reveals a noticeable expansion of the orange area on 04/07. This suggests that the anticyclone at this location is indeed larger in spatial extent.

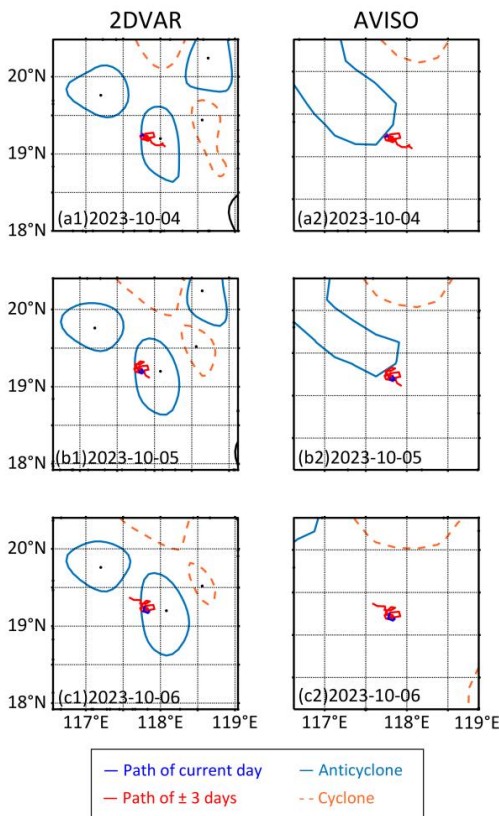

**Figure 5. Trajectories of drifting buoys with 2DVAR (left column) and AVISO (right column) from 2023/10/04 to 10/06. The light blue solid (orange dashed) line, dark blue line segment, and red line segment represent the anticyclonic (cyclonic) eddies, the trajectory for the current day, and the trajectories for the three days before and after, respectively.**

An example of the GDP drifter trajectories compared to the corresponding merged-map eddies is presented in Fig. 5. During October 4–6, 2023, an anticyclonic eddy was identified in the 2DVAR results at 118°E, 19°N, with a total of seven days of drifter buoy trajectory segments within its boundaries. This indicates that the buoy was continuously rotating at a relatively fixed position for approximately seven days. In contrast, AVISO failed to captured this anticyclonic eddy at the same location but instead identified a larger-scale anticyclonic eddy in the vicinity. This discrepancy highlights the accuracy and reliability of the mesoscale eddy detected by 2DVAR, despite a minor positional deviation.

Similar to Figures 3 and 4, we have endeavored to select cases of eddy detection that are representative for both products. However, due to the limited spatial distribution and low observational frequency of the drifter data, the number of valid eddy detection cases available for analysis is highly constrained. In other instances, the differences between 2DVAR and AVISO in comparison with the drifter data are not significant (either both match the drifter trajectories or neither does). The case involving drifter 300534064134530 stands out as the only example demonstrating a clear and favorable comparison.

## 3.3 Eddy Distribution and Characteristics

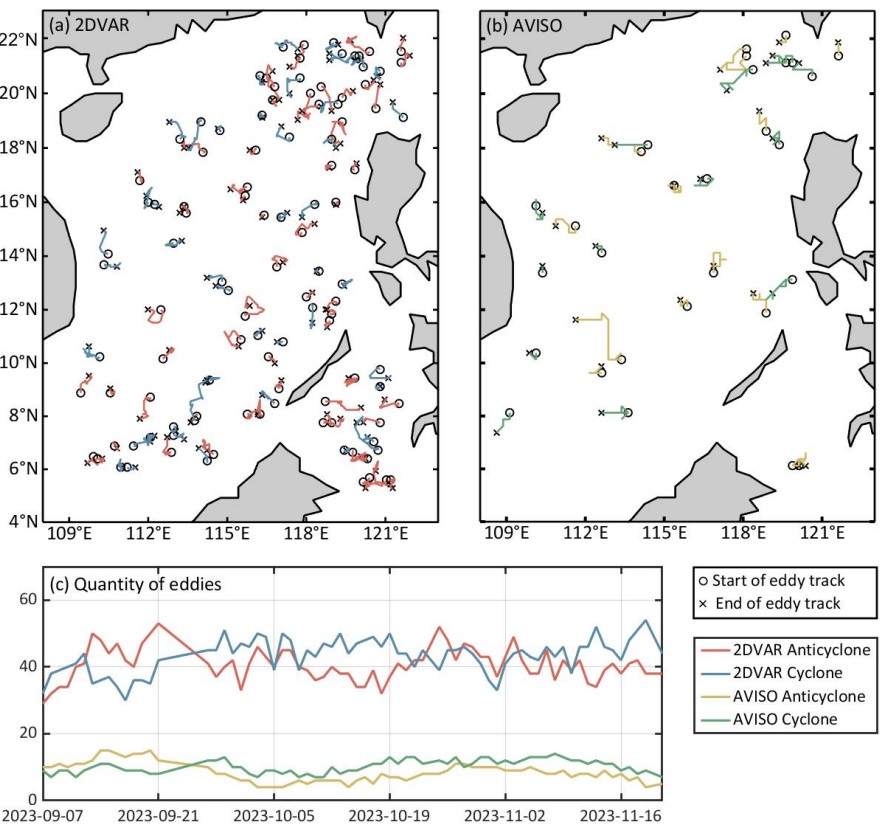

**Figure 6. Eddies and tracks identified during the SWOT science phase (cycle number from 3 to 6, time period from 2023/09/06 to 11/21) in (a) 2DVAR and (b) AVISO merged maps. The red (yellow) and blue (green) lines represent the anticyclone and cyclone tracks in the 2DVAR (AVISO) merged maps, respectively. The black circles and crosses indicate the start and end positions of the eddies, respectively. (c) The number of eddies over time. The red (yellow) and blue (green) lines represent the daily count of anticyclones and cyclones, respectively, as identified in the 2DVAR (AVISO) merged maps.**

In this section, we utilized data spanning several times the satellite observation period within the scientific phase, ensuring comprehensive coverage of the South China Sea region. Based on this dataset, we generated a distribution map of eddy characteristics. The eddy distributions and their tracks were mapped in Fig. 6 to provide a clearer representation of the identified eddies.

The total number of anticyclones identified in the ADT merged maps was higher than that of cyclones, and the eddy tracks exhibited a northeast-southwest propagation and distribution pattern (Figs. 6a–b). This quantitative relationship and distribution pattern are consistent between the two merged maps and are thought to be related to the path of eddies detached from the Kuroshio and intruding into the SCS via the Luzon Strait (approximately 121°E, 20°N) (Huang et al., 2017; Jia and Chassignet, 2011). However, there was a significant discrepancy in the number of eddies identified by the two merged maps: 2DVAR identified approximately four times as many eddies as AVISO, both in terms of the total number of eddies from

315 September to November and the daily number of eddies (Fig. 6c). Notably, in the western part of the Luzon Strait, around 119°E, 20°N, and below 12°N, 2DVAR identified a significantly greater number of eddies compared to AVISO.

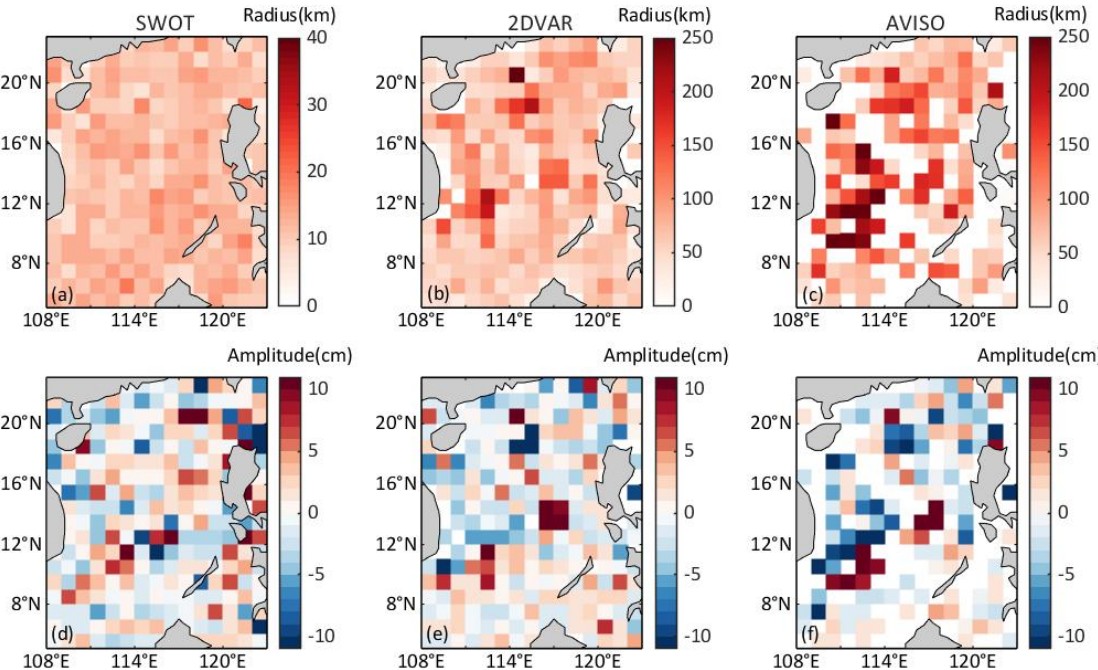

**Figure 7. Distributions of eddy radius (top) and amplitude (bottom) for SWOT (left column), 2DVAR (middle column), and AVISO (right column) from 2023/09/06 to 11/21 are presented. The colour intensity is proportional to the radius, with darker colours**
**indicating larger radii. Similarly, the colour intensity is proportional to the amplitude, with darker red (blue) indicating larger positive (negative) amplitudes.**

The distributions of radius and amplitude for the eddies from merged maps and SWOT maps are displayed in Fig. 7. The radii of the eddies identified in the 2DVAR and AVISO maps are concentrated in the range of 50–300 km (Figs. 7b–c), while the radii of the SWOT eddies are mostly under 50 km (Fig. 7a). This is partly due to the 120 km-wide observation 325 swath of SWOT, which restricts the capture of eddies larger than 120 km.

Larger mesoscale eddies were captured in the southwestern part of the SCS in the merged maps (Figs. 7b–c), whereas no eddies were observed in the SWOT maps (Fig. 7a). This is because eddies in this region may only exist at scales larger than the SWOT observation swath. Both merged maps captured a considerable number of larger mesoscale eddies with radii exceeding 200 km, which were more uniformly distributed in the northern and southwestern parts of the SCS, attributed to 330 the influence of topographic effects (Su et al., 2020).

However, the radii of 2DVAR eddies are approximately 50–100 km smaller than those of AVISO, and a greater number of fine-scale and mesoscale eddies with radii below 150 km were captured in the central and southern parts of the SCS. This is attributed to the smaller effective resolution of 2DVAR (Liu et al., 2020).

The amplitudes of 2DVAR and AVISO are within ±10 cm, while the amplitudes of SWOT are within ±6 cm, which is 4 cm 335 smaller than those of the merged maps in both positive and negative directions.

# 4 Summary and discussion

This research leverages cutting-edge SWOT data to develop an advanced evaluation framework centred on eddy identification within merged maps, achieving superior validation capabilities. Initially, the identified eddies are meticulously normalized and compared with actual eddies. Subsequently, eddy boundary details of the identified eddies are visually compared with those of the merged maps. Finally, the method is validated through observations, ensuring robustness and reliability. This comprehensive approach provides a rigorous assessment of the authenticity and precision of the merged-map eddies, with a detailed analysis of the evaluation outcomes. The main outcomes were summarized as follows:

- The 2DVAR method has a fine-scale to mesoscale eddy identification ratio that is 2.5 to 4 times higher than that of AVISO and exhibits a 50% improvement in the RMSE of eddy radii compared to AVISO, when validated against SWOT.

- Eddies identified in the 2DVAR demonstrated superior coherence and agreement with SWOT data, especially for fine-scale eddies, compared to AVISO.

The results show that 2DVAR identified a significantly greater number of accurate fine-scale and mesoscale eddies compared to AVISO, which is consistent with earlier evaluations of 2DVAR in terms of error analysis, wave number energy spectrum, effective resolution, and OSSE (Observing System Simulation Experiment) (Archer et al., 2020; Jiang et al., 2022). The effective resolution indicates the minimum spatial scale of signals that the merged maps can theoretically resolve, although it does not necessarily correspond to the actual minimum scale. The average effective resolution of AVISO in the SCS is approximately 150 km, whereas that of 2DVAR is about 80 km. This suggests that, typically, AVISO identifies larger eddies compared to 2DVAR. As a result, AVISO encounters greater limitations in identifying eddies across the mesoscale to fine-scale spectrum. Considering the small error correlation scales in high eddy kinetic energy regions such as the South China Sea and other coastal areas, the rational selection of the merged product's regional configuration is also very important. In terms of the trade-off between result performance, the background field time window for AVISO is selected as a multi-year average field, which leads to an increase in the scale of background error signals, and the scale of signals will be amplified during the mapping process. Therefore, the merged map has difficulty in reconstructing and identifying small-scale processes such as small-scale eddies, and will identify more large and mesoscale eddies. Due to the narrow swath of the SWOT track, the number of large and mesoscale eddies that can be identified by AVISO within this range is limited, which may ultimately lead to the limited number of eddies detected by AVISO. In contrast, 2DVAR adopts a one-day background field time window, and the reduction in background error correlation scales allows for the reconstruction of more fine-scale to mesoscale signals.

The successful matches with SWOT eddies on scales less than 20 km further support the argument that fine-scale to mesoscale eddies may have been overlooked in the merged maps. To be noticed, the results showed in this research should be interpreted as the best-case scenario because the eddy identification used for the 2DVAR, AVISO and SWOT maps was identical, implying that the method is perfect. However, several deficiencies may cause errors, including inappropriate eddy

determinations in daily maps without matching with track results. It is because that the eddies moves as tens of kilometers a day which is almost the same with the SWOT swath, resulting in short-trajectory eddies being incorrectly determined as actual eddies in SWOT or merged maps. Also, ignoration of non-closure of contour lines in SWOT maps might be a deficiency too. Most of the eddy scales in the SWOT maps do not exceed 50 km, which limits their ability to fully represent mesoscales eddies, especially those larger than 50 km. Additionally, despite being accurate in terms of radius scale and boundary details, a significant discrepancy remains in terms of positional deviation, which may result in a false match between merged-map eddies with the actual eddies. To address method deficiencies, one possible avenue for improvement would be to use AI or machine learning algorithms for eddy identification and matching, along with auto tracking algorithms to select eddies that persist over time within a limited swath. Compared to the GDP, the validation capability of SWOT is enhanced in both temporal and spatial aspects. This is attributed to the high cost and sparse distribution of drifter platform observations. In contrast, the CALVAL phase of SWOT provided a robust dataset for studying short scale dynamical structures over time. SWOT has now entered the official operational phase, which means it will no longer provide regionally repetitive data of one-day rapid sampling, and the data from the CALVAL phase have become particularly valuable.

The innovative approach presented in this research optimizes and broadens the applications for SWOT data, marking an advancement in the assessment of dynamic signals at the sea surface, particularly at fine-scale. Traditional nadir altimeters, due to their linear single-point sampling method, lack the capability to identify ocean eddies, which have a two-dimensional structure. Therefore, compared to traditional nadir altimeters, the advantage of SWOT lies in its ability to quantitatively analyze the observable scales of two-dimensional structures, such as eddies, within the study area. Since the SWOT data have proven their ability to represent fine-scale eddies through in-situ calibration experiments (Zhang et al., 2024), they can be used as input information for the 2DVAR merging method in the future. With the release of the latest available data, our research will continue to leverage SWOT for enhancing merging methodologies and validating sea surface dynamical structures. The combination of SWOT and 2DVAR will theoretically help improve the effective resolution and accuracy of the merged maps, and this work can provide technical support for oceanographic understanding and prediction capabilities.

## Code availability

The codes are available from Zenodo (https://doi.org/10.5281/zenodo.13629576).

## Data availability

The 2DVAR data are available from Zenodo (https://doi.org/10.5281/zenodo.11219285). The 1/4°AVISO Reprocessed and Nrt data are available from the Copernicus Marine Service repository (2023a, 2023b). The SWOT Level 3 KaRIn Low-Rate Sea Surface Height Data Product, version 1.0.2, is available at (AVISO/DUACS, 2024). The drifter data were supported by GDP (Elipot et al., 2016).

## Author contribution

XZ and LL conceptualized and designed the methodology; XZ conducted the investigation; JF developed the software; ZL and ZW curated the data and performed formal analysis; ZZ provided resources; XJ supervised the project; ZD managed its administration; FX acquired funding; XZ wrote the original draft; and all authors reviewed and edited the manuscript.

## Competing interests

The authors declare that they have no conflict of interest.

## Acknowledgement

This study was supported by the National Natural Science Foundation of China under Grant No. 42192552 and the National Key Research and Development Program of China under Grant No. 2019YFC1510001.

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
