# Peer review of "Advances in Surface Water and Ocean Topography for Fine-Scale Eddy Identification from Altimeter Sea Surface Height Merging Maps in the South China Sea"

_EGUsphere, 2024_

## Referee Comment (RC1)

Review of manuscript egusphere-2024-2773 entitled " Advances in Surface Water and Ocean Topography for Fine-scale Eddy Identification from Altimeter Sea Surface Height Merging Maps"

**Main comment:**

Within the manuscript the authors use the new SWOT high-resolution data to detect eddies in the South China Sea and compare the results with eddy identification from the conventional altimetry AVISO product and a merged product using a 2DVAR method. Their analysis show that the 2DVAR method exhibit better reconstruction of the fine-scale ocean dynamics compared to conventional altimetry-derived product.

The paper is well-written and well structured however the analysis is restricted only at showing that the 2DVAR is comparing better with SWOT than the conventional altimetry product. A result that is interesting and worth proving it but as far as I am concerned the publication lacks of analysis on the improvements brought by the new 2DVAR method concerning scientific questions. What do we learn, using this new data, in terms of fine-scale processes in this dynamical region ? Secondly, the publication will benefit from more robust statistical quantifications of the improvement of the new method. The authors only rely on statistics over the eddy radius and boundary detection. More quantification on the performance of the new 2DVAR method should be added (see comments below).

Lastly I feel like the title may be misleading. As is I was expecting some new findings on the fine-scale eddy identification in the South China Sea while the analysis stick with a comparison between higher resolution observations of the sea surface and a new methodology (2DVAR). Maybe the authors should consider to modify slightly the title if no further analysis are added to the manuscript.

To summarize, in its current form the paper is very interesting and brings some relevant evidence to validate the 2DVAR method but needs further analysis before it can be published. Therefore I would recommend to publish the manuscript after some major revision. Please find in the following my detailed comments.

**Major comments:**

1)  The authors are using SWOT data from a pretty old version now (v0.3 on L 100). Two improved versions have been released for both CALVAL and Science phase: v1.0 or v1.0.2. I want to bring the authors attention on the fact that a new improved version of SWOT data (v2.0) is going to be released around December 2024. This new version (v2.0) is of particular interest because it is going to include a MDT correction that will improved SSH (and therefore ADT) estimates of about 5 cm. I believe this correction is of great interest for this study. I would thus recommend to re run the SWOT analysis using the latest version v2.0 (when available) or at the very least to use v1.0.2.

2) Section 2.3: I have several questions about the methodology to compare the performances of 2DVAR and AVISO with SWOT data:
-  First the authors, only compare the eddies that are detected by conventional altimetry and SWOT. But I think a real and very interesting point would also be to quantify how much of the eddies observed by SWOT are NOT detected by conventional altimetry and 2DVAR ? And probably to compute a metric showing the better performance of 2DVAR in identifying fine scale features observed by SWOT and not by the conventional product.
-  I understand how the method can identify eddy center and boundary (L 126-140) with a gridded product such as AVISO or 2DVAR but I do not get how the author can estimate eddy center/boundary using SWOT data considering that they only exhibit data on the swaths and that eddies can be partly outside of the observational bands of SWOT. Please provide more information.
-  The authors normalize the position of eddy center and boundary by using SWOT data. This technique smooth the differences between gridded product and SWOT data based eddy identification. Why would

one want to do that if the point is to compare the performances of the different gridded product to adequately represent high resolution observation from SWOT ?

3) Section 3.1: The analysis of Fig.2 needs more explanations as it is not always straightforward to understand what is represented on Fig. 2. For example the colorbar ranges are different in all plots, I don't understand if the colors represent a number of eddies or the eddy radius ?

4) Section 3.2 "Eddy boundaries verification in space and time": The authors show one example of GDP drifter comparison with 2DVAR and AVISO. It is a good illustration however I think the authors should provide more statistics with a systematic comparison of drifters during the science phase to clearly demonstrate the better accuracy between in situ and 2DVAR compared to AVISO.

5) Section 3.3: These results seem inherently due to the different products resolution, so not really surprising. I think one of the main result of the study, listed also on L282, is mainly due to the fact that SWOT do not capture features > 120 km due to the swaths limitation, as stated by the authors on L 295. So I am not sure this result should appear like a main point of the paper since it is something that is limited by the data.

**Minor comments:**

L 33: "a kind of", I would state "is an estimate of sea surface height (SSH) above geoid."

L 37: Conventional altimetry does not really provide "high resolution" mapping since the resolved processes are mesoscale features (75-100 km). I would remove "high resolution" from the sentence and add "…tracking large and mesoscale ocean dynamic signals (Chelton…".

L 42: Worth mentioning the interpolation techniques (Pujol et al., 2016).

L 67-69: Please define the period for CALVAL and Science Phase for SWOT mission. I believe not all readers are aware of that (CALVAL: Mar-July 2023; SP: from August/September 2023). It seems to be done on Lines 106-107 but should appear when the terms are first used.

L 75: "fine scales" are not defined previously in the text. Please give details (maybe in the introduction?). Are the authors referring to submesoscales ? Or more classically to the transition scale between meso to submesoscales ?

L 87-88: There are different types of eddy identification methods: physical, geometric, Lagrangian or hybrid based methods. The authors need to provide some references here (among MCWilliams, 1990; Okubo, 1970; Weiss, 1991; Chelton et al., 2011; Sadarjoen and Post, 2000; Mkhinini et al., 2014; Laxenaire et al., 2018 …).

L 94: Is it based on Chelton et al., 2011 method ?

L 95-97: I guess the steps and amplitude in mm and cm are referring to distances on the maps. Can you please give the actual distance in kilometer to make a connection with data resolution ?

L 109-110: ".. and lack of interest in traditional technology". I would remove this comment since nadir data are still being used and methodologies developed to assess fine scale improvements near the coasts (Birol et al., 2021).

L 150: Did the authors performed any kind of treatment on the drifter trajectories (GDP)? For example, are the inertial oscillations removed from the trajectories before comparison ?

L 158: Depending on the authors definition of submesoscale, mesoscale and fine-scale (see previous comment) I would rather classify as "submsesoscale (Fig. 2a), fine-scale (Fig. 2b) and mesoscale (Fig. 2c)".

L 166-169: I do not understand the reasoning here, please detail and rephrase.

L 175: I do not think that AVISO can catch eddies down to 15 km. Conventional altimetry typically see mesoscale features with diameter of about (75-100 km).

L 188-189: I think the maximum amplitude of SWOT eddies might change when the 5 cm bias will be corrected from MDT in the new SWOT data version 2.0… Worth checking out when the new data are available!

L 200: "The coloured slices are .." ?

Fig. 3: SWOT data seems to be interpolated between swaths. Please provide information. Please specify in the caption that solid and dashed lines represent the contours of eddies as detected by 2DVAR and AVISO.

L 217: Do the authors have any clue why the contour of the 16.2N anticyclone is very different on the 04/07 while the contours are pretty consistent in the other days ?

L 218: I would change "high agreement" by "good agreement".

L 224: It is not really surprising that AVISO does not identify 50 km eddies.

L 237: The section is not correctly numbered, it should be Section 3.3.

L 250: add space between "eddies" and "identified".

Fig. 6-7: Please provide the period range for the analysis during the Science phase (also the number of SWOT passes used ?).

**References:**

Pujol, M. I., Faugère, Y., Taburet, G., Dupuy, S., Pelloquin, C., Ablain, M., & Picot, N. (2016). DUACS DT2014: the new multi-mission altimeter data set reprocessed over 20 years. *Ocean Science*, *12*(5), 1067-1090.

McWilliams, J. C. (1990). The vortices of two-dimensional turbulence. *Journal of Fluid mechanics*, *219*, 361-385.

Okubo, A. (1970, June). Horizontal dispersion of floatable particles in the vicinity of velocity singularities such as convergences. In *Deep sea research and oceanographic abstracts* (Vol. 17, No. 3, pp. 445-454). Elsevier.

Weiss, J. (1991). The dynamics of enstrophy transfer in two-dimensional hydrodynamics. *Physica D: Nonlinear Phenomena*, *48*(2-3), 273-294.

Chelton, D. B., Schlax, M. G., & Samelson, R. M. (2011). Global observations of nonlinear mesoscale eddies. *Progress in oceanography*, *91*(2), 167-216.

Sadarjoen, I. A., & Post, F. H. (2000). Detection, quantification, and tracking of vortices using streamline geometry. *Computers & Graphics*, *24*(3), 333-341.

Mkhinini, N., Coimbra, A. L. S., Stegner, A., Arsouze, T., Taupier-Letage, I., & Béranger, K. (2014). Long-lived mesoscale eddies in the eastern Mediterranean Sea: Analysis of 20 years of AVISO geostrophic velocities. *Journal of Geophysical Research: Oceans*, *119*(12), 8603-8626.

Laxenaire, R., Speich, S., Blanke, B., Chaigneau, A., Pegliasco, C., & Stegner, A. (2018). Anticyclonic eddies connecting the western boundaries of Indian and Atlantic Oceans. *Journal of Geophysical Research: Oceans*, *123*(11), 7651-7677.

Birol, F., Léger, F., Passaro, M., Cazenave, A., Niño, F., Calafat, F. M., ... & Benveniste, J. (2021). The X-TRACK/ALES multi-mission processing system: New advances in altimetry towards the coast. *Advances in Space Research*, *67*(8), 2398-2415.

---

## Author Comment (AC1)

**Reply on RC1**

*Thank you for your comments concerning our manuscript. Those comments are all valuable and very helpful for revising and improving our paper. We have studied comments carefully and have made correction which we hope meet with approval. The main corrections in the paper and the responses to your comments are highlighted in blue and are as follows:*

**Major comments:**

1) The authors are using SWOT data from a pretty old version now (v0.3 on L 100). Two improved versions have been released for both CALVAL and Science phase: v1.0 or v1.0.2. I want to bring the authors attention on the fact that a new improved version of SWOT data (v2.0) is going to be released around December 2024. This new version (v2.0) is of particular interest because it is going to include a MDT correction that will improved SSH (and therefore ADT) estimates of about 5 cm. I believe this correction is of great interest for this study. I would thus recommend to re run the SWOT analysis using the latest version v2.0 (when available) or at the very least to use v1.0.2.

**Response:** Thank you very much for your valuable suggestion on the data version update. Regarding the issue of data quality correction in the new version (2.0) of SWOT that you mentioned, I had already raised concerns about the nadir data quality of the SWOT mission to the team leader, Dr. Morrow, in April 2024. I pointed out that in the same version of the SWOT product, there were significant and unreasonable deviations between the KaRIn data and the nadir data at similar locations. In response, Morrow indicated that this discrepancy was due to the nadir data using an earlier version of the Mean Dynamic Topography (MDT) (i.e., the 2018 version), which was estimated to have a bias of 5 cm (see the email reply content below in S1). The 5-cm error improvement in the MDT estimation mentioned in the SWOT v2.0 version is precisely aimed at addressing this issue. Since we identified this problem early on, in the actual implementation of this study, we had already excluded all traditional nadir observational data and conducted experiments using only KaRIn data. Therefore, the update of the MDT in version 2.0 will not affect our experiments. Additionally, as of January 9, 2025 (our last check date), the v2.0 data had not yet been released on the official website. Consequently, we proceeded to download the v1.0.2 version to replace the previous v0.3 version data and re-conducted all SWOT-related experiments using the new data. Please review the relative changes illustrated in the figures in the revised manuscript.

[Figure]

Rosemary Morrow UT3  2024-04-08 19:10
发至 zxy777；抄送 Pujol Marie-isabelle、Dibarboure Gerald、J. Tom Farrar

Dear Xiaoya

Thank you for pointing this anomaly out to us. We have been concentrating so much on the new KaRIn processing, that the differences with the nadir have taken second priority

I checked with the processing centres, and this is why ...

1. The MDT used in the KaRIn product is the MDT CNES_CLS_2022
2. The MDT in the nadir product is the MDT CNES_CLS_2018, biased by 5 cm.

The 5 cm bias used in the nadir product  was set  in 2014, to maintain the continuity in the 30-year time series of ADT measurements for all radar altimetry missions , following a change in the 20-year reference period for the MDT.

So there is this difference in the two MDTs between the 2 products. In the next version of the L3 products, they aim to fix this MDT difference, and only use the more recent product in both the nadir and KaRIn.

We will keep the Science Team informed on updates with the new processing steps, in the meantime you should explain and adjust for this 5 cm nadir bias. And thank you for your helpful return on this subject

sincerely yours

Rosemary Morrow

S1 Email from Morrow

2) Section 2.3: I have several questions about the methodology to compare the performances of 2DVAR and AVISO with SWOT data:

- First the authors, only compare the eddies that are detected by conventional altimetry and SWOT. But I think a real and very interesting point would also be to quantify how much of the eddies observed by SWOT are NOT detected by conventional altimetry and 2DVAR ? And probably to compute a metric showing the better performance of 2DVAR in identifying fine scale features observed by SWOT and not by the conventional product.

**Response:** Thank you for your nice suggestion about the metric of eddy identification. To quantify the number of eddies observed by SWOT but not captured by the merged map, we have included the total number of identified eddies in the upper-left corner of Figure 3 and simultaneously updated the eddy identification ratio in Table 1. We have also added the following description in the text.

**Line 209-214:** We also calculated the eddy identification ratio based on the eddy quantity in the merged map compared to that in the SWOT data. The results demonstrate that as the SWOT observation radius increases, the eddy identification ratio of the 2DVAR method rises from 25% to 40%, while the identification ratio of AVISO remains relatively stable at around 11%. This leads to a significant increase in the gap between the two methods, from 2.5 times to 4 times. This contrast highlights the superior performance of the 2DVAR method in detecting eddies using SWOT data, especially in capturing fine-scale to mesoscale features that AVISO may miss.

- I understand how the method can identify eddy center and boundary (L 126-140) with a gridded product such as AVISO or 2DVAR but I do not get how the author can estimate eddy center/boundary using SWOT data considering that they only exhibit data on the swaths and that eddies can be partly outside of the observational bands of SWOT. Please provide more information.

**Response:** We apologize for not clarifying this part. In fact, the eddy identification method used in the SWOT data is consistent with the method used in the merged map product. However, this method does have certain limitations in identifying eddies within the SWOT swath, specifically those eddies that are only partially included in

the SWOT swath and cannot be identified by the closed-contour method. These eddies are indeed not identified and tracked. Nevertheless, the primary focus of this study is to use the SWOT data to validate the capabilities of the merged map, rather than to focus on the number or quality of eddies that SWOT itself can identify. Therefore, we believe that this eddy identification method is sufficient to support the research presented in this paper.

**Line 119-120:** It is worth noting that, due to SWOT observation data limitations, we are currently unable to identify eddies located at the edges of the swath or outside the swath.

- The authors normalize the position of eddy center and boundary by using SWOT data. This technique smooth the differences between gridded product and SWOT data based eddy identification. Why would one want to do that if the point is to compare the performances of the different gridded product to adequately represent high resolution observation from SWOT ?

**Response:** Your question is excellent, and I apologize for not explaining the purpose of this operation clearly. In fact, the normalization process applied in this study is used for all eddies identified by SWOT. The purpose is to smooth out the differences among various SWOT eddies to facilitate a highly statistical representation of all SWOT eddies (as shown by the black dashed circles in Figure 2). Meanwhile, we applied the same normalization coefficients derived from the SWOT eddies to the eddies in the two merged maps (2DVAR and AVISO) and performed a linear scaling proportional to the SWOT eddies, preserving the relative differences between the merged maps and SWOT. This normalization and proportional scaling allow us to statistically synthesize the differences of tens of thousands of eddies into a single standard grid composite map, which is an effective way to depict the statistical characteristics of the radius differences between the merged maps and SWOT eddies (i.e., the differences between the colored parts and the dashed circles in Figure 2). We have made corresponding modifications in the original text regarding this issue.

**Line 147-151:** The operation aims to standardize SWOT eddies of diverse sizes and proportionally scale the eddies in the 2DVAR and AVISO merged maps relative to those in SWOT, thereby retaining the relative differences between the merged maps and SWOT. This normalization and proportional scaling allow the statistical synthesis of differences among thousands of eddies into a single standard grid composite map, effectively characterizing the radius discrepancies between the merged maps and SWOT eddies.

3) Section 3.1: The analysis of Fig.2 needs more explanations as it is not always straightforward to understand what is represented on Fig. 2. For example the colorbar ranges are different in all plots, I don't understand if the colors represent a number of eddies or the eddy radius ?

**Response:** Thank you very much for pointing this out. The color bar in Figure 2 represents the eddy count, and a corresponding description has been added to the caption.

**Line 230-233:** Composite maps of normalized eddy identified from 2DVAR (left column) and AVISO (right column) merged maps, the color on the grid points shows the density of covered eddies, with the higher the density of the grid points, the darker the orange. The dashed circles mark the normalized SWOT eddies with a radius of less than 10 km (a), 10 to 20 km (b), and more than 20 km (c). The total number of eddies detected by each map is on the left up corner.

4) Section 3.2 "Eddy boundaries verification in space and time": The authors show one example of GDP drifter comparison with 2DVAR and AVISO. It is a good illustration however I think the authors should provide more statistics with a systematic comparison of drifters during the science phase to clearly demonstrate the better accuracy between in situ and 2DVAR compared to AVISO.

**Response:** Thank you very much for your valuable suggestions. This study focuses primarily on the enhanced and more accurate data that SWOT can provide compared to traditional in situ observations for validating the quality of merged maps. Therefore, Figure 5 in Section 3.2 is designed to illustrate the efficiency difference between the latest altimetric techniques and traditional altimetry in examining eddies in merged maps. On the other hand, due to the limited and concentrated distribution of drifter data within the study area, and considering that velocities are derived from sea surface height inversions, which have higher sampling requirements and thus introduce additional errors (Pascual, 2007), we did not consider using statistical data for validation.

5) Section 3.3: These results seem inherently due to the different products resolution, so not really surprising. I think one of the main result of the study, listed also on L282, is mainly due to the fact that SWOT do not capture features > 120 km due to the swaths limitation, as stated by the authors on L 295. So I am not sure this result should appear like a main point of the paper since it is something that is limited by the data.

**Response:** Thank you for your valuable feedback on the conclusion. If the conclusion you are referring to is: "Eddies identified in the 2DVAR demonstrated superior coherence and agreement with SWOT data, especially for fine-scale eddies, compared to AVISO," then we appreciate the opportunity to clarify further. Indeed, we have already unified the grid resolution of the different data sources to 1/12° prior to eddy identification. Despite this uniform grid resolution, differences in eddy identification persist. This suggests that the source of these differences does not stem from data resolution but rather from the differences in the fusion methods used.

**Line 121-122:** To avoid the influence of grid resolution, different merged maps were interpolated to a high-resolution grid with the same resolution (1/12°).

**Minor comments:**

L 33:"a kind of", I would state "is an estimate of sea surface height (SSH) above geoid."

**Response:** Thank you for your correction. The original text has been revised.

**Line 33:** These processes are primarily revealed through the absolute dynamic topography (ADT), which is an estimate of sea surface height (SSH) above the geoid.

L 37:Conventional altimetry does not really provide "high resolution" mapping since the resolved processes are mesoscale features (75-100 km). I would remove "high resolution" from the sentence and add "…tracking large and mesoscale ocean dynamic signals (Chelton…".

**Response:** Thank you for your correction. The original text has been revised.

**Line 38:** Global satellite altimeters offer systematic ADT measurements and mapping of ocean topography, currently providing the most effective data for detecting and tracking large and mesoscale ocean dynamic signals (Chelton et al., 2007; Mason et al., 2014; Zhang et al., 2023).

L 42:Worth mentioning the interpolation techniques (Pujol et al., 2016).

**Response:** Thank you for your suggestions on the literature review. We have added a description of the interpolation methods used in the merged maps.

**Line 42-45:** The main techniques of data assimilation typically include the homogenization and cross-calibration of multi-source altimetry data, continuous calibration of reference orbits, cross-calibration between altimeters, long-wavelength error correction, and error budget modeling. Finally, optimal interpolation is used for gridding to generate daily gridded products and derived products (Pujol et al., 2016).

L 67-69:Please define the period for CALVAL and Science Phase for SWOT mission. I believe not all readers are aware of that (CALVAL: Mar-July 2023; SP: from August/September 2023). It seems to be done on Lines 106-107 but should appear when the terms are first used.

**Response:** Thank you very much for pointing out the issues. We have moved all the

introductions related to SWOT data to the first paragraph of Section 2.1 and have correspondingly adjusted the order of the figures. The following are the modifications related to SWOT.

**Line 69-77:** The SWOT mission consists of two phases: the science phase, which conducted 21-day repeat sampling from September 7th 2023 to November 21st 2023, and the calibration and validation phase (CALVAL), which performed 1-day rapid sampling from April 1st 2023 to July 31st 2023. The CALVAL phase data were used exclusively in the second part of Section 3.2 to analyse the temporal evolution of eddies. In contrast, the science phase data were the primary datasets for examining the spatial dynamic structures and for performing statistical analyses of eddy characteristics. Although the CALVAL phase sampled a limited sea surface area due to its fixed rapid-sampling orbit, this orbit covered part of the South China Sea (SCS) and facilitated the capture of time-evolving fine-scale eddy structures in the SCS. The nadir observation points, located between two slices of KaRIn observations (Fig. 1(a)), were excluded in both phases due to their high error rates and our focus on the advanced KaRIn technology.

L 75:"fine scales" are not defined previously in the text. Please give details (maybe in the introduction?). Are the authors referring to submesoscales ? Or more classically to the transition scale between meso to submesoscales ?

**Response:** Thank you for raising the issue. We have now defined "fine scales" in the Introduction section.

**Line 59-60:** Fine-scale ocean processes are characterized by spatial variability of 1 to 100 kilometres and temporal variability of days to months (Lévy et al., 2024).

L 87-88:There are different types of eddy identification methods: physical, geometric, Lagrangian or hybrid based methods. The authors need to provide some references here (among MCWilliams, 1990; Okubo, 1970; Weiss, 1991; Chelton et al., 2011; Sadarjoen and Post, 2000; Mkhinini et al., 2014; Laxenaire et al., 2018 …).

**Response:** Thank you for your correction. We have improved the literature review of the eddy identification methods, and the references have been added.

**Line 111-115:** Currently, the main methods for eddy identification based on satellite altimeters include the Okubo-Weiss (OW) parameter from the velocity field method, the curvature center method, the surrounding angle method, the local extreme of sea surface topography method, the local and normalized angular momentum method, and the Lagrangian coherent structure (LCS) method (Chelton et al., 2011; Laxenaire et al., 2018; Mcwilliams, 1990; Mkhinini et al., 2014; Okubo, 1970; Sadarjoen and and, 2000; Weiss, 1991).

L 94: Is it based on Chelton et al., 2011 method ?

**Response:** Yes, the eddy identification method used in this study is based on the method described in the article by Ni, which in turn is derived from the method proposed by Chelton et al. (2011). Therefore, we have updated the references to cite the primary source by Chelton et al. (2011) .

**Line 116-118:** Therefore, this research employed a sea surface topography method based on contour analysis for eddy identification in 2DVAR and AVISO ADT merged maps, as well as in SWOT maps (Chelton et al., 2011).

L 95-97: I guess the steps and amplitude in mm and cm are referring to distances on the maps. Can you please give the actual distance in kilometer to make a connection with data resolution ?

**Response:** I apologize for causing confusion with my wording. Actually, the step in mm refers to the elevation difference (1 mm) between contour lines set for retrieving the contour lines in the ADT maps, not the real distance on the map. The amplitude in cm refers to the difference in ADT between the eddy center and the boundary contour line in the possible eddies that had been identified in the previous steps. I will revise it to a more reader-friendly expression.

**Line 122-127:** The outermost circle of the closed contours with 1mm step in ADT difference containing the unique centre was recognized as the 'quasi-eddy edge', and only a minimum of three points were retained. Each quasi-eddy edge was then contracted inward until it corresponded to a single centre. Lastly, the geometric centre of the innermost circle of the closed contours was identified as the eddy centre. This process allowed the determination of the eddy boundary, type, radius, and amplitude. All possible eddies with an difference in ADT between the eddy center and the boundary contour line of less than $\pm 2$ cm were excluded from further analysis.

L 109-110: ".. and lack of interest in traditional technology". I would remove this comment since nadir data are still being used and methodologies developed to assess fine scale improvements near the coasts (Birol et al., 2021).

**Response:** I apologize for any confusion caused. The original wording was indeed unclear. What I meant to convey is that, due to the instability of the Nadir-type data in the SWOT dataset and its irrelevance to the research interests of this study, these data were excluded from our analysis prior to conducting the research.

**Line 75-77:** The nadir observation points, located between two slices of KaRIn observations (Fig. 1(a)), were excluded in both phases due to their high error rates and our focus on the advanced KaRIn technology.

L 150: Did the authors performed any kind of treatment on the drifter trajectories (GDP)? For example, are the inertial oscillations removed from the trajectories before comparison ?

**Response:** Thank you for raising the technical issue. Following your suggestion, I applied a low-pass filter with a 3-day time window to eliminate high-frequency inertial oscillation signals. Comparing the plot after removal (S2) with the plot before removal (S3), the dynamic effect of the drifter orbiting within the eddy is significantly reduced. Therefore, I believe that not applying any additional processing better illustrates the relationship between the drifter rotating along the eddy boundary.

[Figure]

S2 GDP paths after filtering.          S3 GDP paths before filtering.

L 158: Depending on the authors definition of submesoscale, mesoscale and fine-scale (see previous comment) I would rather classify as " submsesoscale (Fig. 2a), fine-scale (Fig. 2b) and mesoscale (Fig. 2c)".

**Response:** Thank you very much for your suggestion. We have revised the classification as per your advice, changing the middle category to submesoscale. Since fine-scale encompasses the smallest eddy identification scales (1-100 km), we have categorized the first class as fine-scale.

**Line 184-186:** All identified eddies were categorized into three groups based on the radii of SWOT eddies: those with radii below 10 km, between 10 km and 20 km, and exceeding 20 km were classified as fine-scale (Fig. 2a), submesoscale (Fig. 2b), and mesoscale (Fig. 2c) eddies, respectively.

L 166-169: I do not understand the reasoning here, please detail and rephrase.

**Response:** Thank you for your question. I apologize for the lack of clarity in my explanation. The reasoning in this paragraph is based on the comparison between the eddy boundaries in different subplots and the normalized SWOT eddy boundaries (marked by dashed circles). I will revise the sentence to make it clearer and easier to understand.

**Line 194-201:** Since the same colorbar is applied to the eddy ensembles of the two merged maps for the same scale range, the distribution and concentration of eddies can be judged by the intensity of the colors. It is evident that the area of the colored region outside the normalized SWOT eddies (marked by black dashed circles) in the 2DVAR merged map is smaller especially in Fig. 2b and 2c. Meanwhile, despite the number of eddies identified by 2DVAR being 2 to 3 times that of AVISO, the color outside the normalized eddy circles remains a light shade of purple (means no more than 10 eddies). These results suggest that the concentration of 2DVAR eddies within the normalized SWOT eddies is higher, and the eddy boundaries maintain a higher degree of consistency.

L 175: I do not think that AVISO can catch eddies down to 15 km. Conventional altimetry typically see mesoscale features with diameter of about (75-100 km).

**Response:** Thank you for your insightful comments. We are very happy to further discuss with you the issue of whether traditional altimetry methods can identify eddies smaller than 75 km. Although the data providers do not explicitly state that they can, I found a study that identified eddies as small as 19 km using data with a 1/8° grid resolution (S4), and half of the identified eddies had radii smaller than 30 km (Wang, 2021) . Therefore, I believe it is possible for us to identify smaller eddies as well.

[Figure]

**Figure 4.** Eddy radius distribution maps and radius histogram of the Mediterranean Sea: (**a**) 1/4°; (**b**) 1/8° all-sat; (**c**) 1/8° two-sat. The colored squares represent the range of eddy radii, as shown in the legend. The histogram (**d**) represents the relationship between eddy radii and number in the different datasets, where the black line represents the eddies identified using the 1/4° data, and the magenta (desert blue) line represents eddies identified using the 1/8° all-sat (two-sat) data.

S4 Eddy radius distribution and radius histogram. (Wang, 2021)

L 188-189: I think the maximum amplitude of SWOT eddies might change when the 5 cm bias will be corrected from MDT in the new SWOT data version 2.0⋯ Worth checking out when the new data are available!

**Response:** Thank you for your valuable suggestions. Since this study did not utilize the Nadir data based on the older version of MDT, which is known to have significant errors, the new version is unlikely to have a substantial impact on the data (please refer to the response to the first question for details).

L 200: "The coloured slices are .." ?

**Response:** Thank you for pointing out the issue. What we intended to convey was "colored points in the figure." We have revised the sentence to express it more clearly and accurately.

**Line 238-239:** The coloured points in the figure showed the Level-3 ADT observations directly from the KaRIn measurement on SWOT satellite.

Fig. 3: SWOT data seems to be interpolated between swaths. Please provide information. Please specify in the caption that solid and dashed lines represent the contours of eddies as detected by 2DVAR and AVISO.

**Response:** Thank you for your questions and suggestions. The SWOT data have been down-sampled through a 1/12° grid average to align with the merged products and to fill the gap between the two KaRIn instrument measurement swaths caused by the removal of Nadir data. The caption of Figure 3 has been corrected to indicate that the lines represent the eddy contours detected by 2DVAR and AVISO, while the colors indicate the SWOT height data.

**Line 249-250:** Figure 3. The ADT observation data of SWOT with the eddies detected by SWOT (in red), 2DVAR (in black, a1-c1) and AVISO (in black, a2-c2). The solid line represents the anticyclonic eddies and dashed line represents the cyclonic eddies.

L 217: Do the authors have any clue why the contour of the 16.2N anticyclone is very different on the 04/07 while the contours are pretty consistent in the other days ?

**Response:** Thank you for raising this question; it is highly relevant. Both the 2DVAR and AVISO products exhibit inconsistent identification of the anticyclone at 16.2°N, with fluctuations in size. This suggests that the merged maps derived from both methods inadequately reconstruct sea surface height at this location during the specified time period, leading to errors such as misrepresenting a small eddy as a larger one. Additionally, in the underlying SWOT data, although it fails to identify an eddy that is only partially within the swath, the color map indicates a noticeable expansion of the orange area on 04/07. This implies that the anticyclone at this

location should indeed be larger. If the SWOT data coverage were more extensive, I believe we would observe a similar sudden enlargement of the eddy at 16.2°N on 04/07, akin to what is seen in the 2DVAR product.

**Line 260-265:** To be noticed, both the 2DVAR and AVISO products exhibit inconsistent identification of the anticyclone at 16.2°N, with fluctuations in size. This suggests that the merged maps derived from both methods inadequately reconstruct sea surface height at this location during the specified time period, leading to errors such as misrepresenting a small eddy as a larger one. Additionally, in the underlying SWOT data, although it fails to identify an eddy that is only partially within the swath, the color map indicates a noticeable expansion of the orange area on 04/07. This implies that the anticyclone at this location should indeed be larger.

L 218: I would change "high agreement" by "good agreement".

**Response:** Thank you for your correction. The text has been revised.

**Line 258-259:** These anticyclones were in good agreement with the colour boundaries in the bottom SWOT data.

L 224: It is not really surprising that AVISO does not identify 50 km eddies.

**Response:** Thank you for raising the question. Given that previous studies have demonstrated the capability of traditional AVISO products to identify eddies smaller than 20 km (Wang, 2021), the inability to detect eddies of 50 km remains a noteworthy issue. Moreover, considering that both 2DVAR and AVISO are traditional altimetry data merged fields based on the Optimal Interpolation (OI) principle, the fact that 2DVAR can identify eddies of 50 km while AVISO cannot is a significant finding. This may suggest inherent differences between different merged field products.

L 237: The section is not correctly numbered, it should be Section 3.3.

**Response:** Thank you for your correction. The numbering in Section 3.3 has been revised.

L 250: add space between "eddies" and "identified".

**Response:** Thank you for pointing out the typographical error. The correction has been made.

**Line 295-297:** However, there was a significant discrepancy in the number of eddies identified by the two merged maps: 2DVAR identified approximately four times as many eddies as AVISO, both in terms of the total number of eddies from September to

November and the daily number of eddies (Fig. 6c).

Fig. 6-7: Please provide the period range for the analysis during the Science phase (also the number of SWOT passes used ?)

**Response:** I apologize for not providing this information. The time range for the science phase is from September 6, 2023, to November 21, 2023, and the pass numbers are 003-006.

**Line 284-288:** Figure 6. Eddies and tracks identified during the SWOT science phase (cycle number from 3 to 6, time period from 2023/09/06 to 11/21) in (a) 2DVAR and (b) AVISO merged maps. The red (yellow) and blue (green) lines represent the anticyclone and cyclone tracks in the 2DVAR (AVISO) merged maps, respectively. The black circles and crosses indicate the start and end positions of the eddies, respectively. (c) The number of eddies over time. The red (yellow) and blue (green) lines represent the daily count of anticyclones and cyclones, respectively, as identified in the 2DVAR (AVISO) merged maps.

**Line 300-303:** Figure 7. Distributions of eddy radius (top) and amplitude (bottom) for SWOT (left column), 2DVAR (middle column), and AVISO (right column) from 2023/09/06 to 11/21 are presented. The colour intensity is proportional to the radius, with darker colours indicating larger radii. Similarly, the colour intensity is proportional to the amplitude, with darker red (blue) indicating larger positive (negative) amplitudes.

---

## Author Comment (AC2)

**Reply on RC2**

*Thank you for your comments concerning our manuscript. Those comments are all valuable and very helpful for revising and improving our paper. We have studied comments carefully and have made correction which we hope meet with approval. The main corrections in the paper and the responses to your comments are highlighted in blue and are as follows:*

Material and method section

To enhance the Materials and Methods section, I would recommend adding several clarifications. First, it would be beneficial to include specific references, such as a DOI, for the gridded sea surface height datasets used in the study. While the datasets are listed in the references, explicitly citing them in this section would improve clarity and accessibility for readers.

**Response:** Thank you for your precious suggestion. The references or DOIs have been added in Section 2.1.

**Line 69-71:** The SWOT mission consists of two phases: the science phase, which conducted 21-day repeat sampling from September 7th 2023 to November 21st 2023, and the calibration and validation phase (CALVAL), which performed 1-day rapid sampling from April 1st 2023 to July 31st 2023 (AVISO/DUACS, 2024).

**Line 78-80:** The first ADT product (Fig. 1b) was produced using a $\pm 11$ days time window Near Real-Time (NRT) Two-Dimensional Variation (2DVAR) method with a $1/12°$ grid resolution (https://doi.org/10.5281/zenodo.11219285).

**Line 83-86:** During the science phase of the SWOT mission, the AVISO merged map Delayed Time (DT) products utilized SWOT data as an input source (Copernicus Marine Service repository, 2023b). To maintain the independence of the datasets, we employed NRT products, which do not include SWOT data (Copernicus Marine Service repository, 2023a).

Additionally, please specify the time period used, as this information is essential for contextualizing the scope of the analysis. I was a bit lost when I read/interpreted the results since I was not sure which CMEMS (DT, NRT) dataset was used.

**Response:** I apologize for previously providing the time period and dataset information in a scattered and vague manner. Below, I have pasted the revisions that organize and summarize the information regarding the data time period and the CMEMS DT/NRT details.

**Line 69-71:** The SWOT mission consists of two phases: the science phase, which conducted 21-day repeat sampling from September 7th 2023 to November 21st 2023, and the calibration and validation phase (CALVAL), which performed 1-day rapid sampling from April 1st 2023 to July 31st 2023 (AVISO/DUACS, 2024).

**Line 83-88:** During the science phase of the SWOT mission, the AVISO merged map Delayed Time (DT) products utilized SWOT data as an input source (Copernicus Marine Service repository, 2023b). To maintain the independence of the datasets, we employed NRT products, which do not include SWOT data (Copernicus Marine Service repository, 2023a). Similarly, during the CALVAL phase of the SWOT mission, we used an older version of the DT products which do not include SWOT data as an input source.

Furthermore, a more detailed explanation of the differences between Near Real-Time (NRT) and Delayed Time (DT) products you are using would be helpful. Clarifying which processing mode (NRT, DT, or both) was used for the 2DVAR mapping would also improve transparency in your methodology and intercomparison.

**Response:** Thank you for your valuable suggestions. I will provide a brief description of the differences in the processing methods for NRT and DT, as well as the processing method for 2DVAR.

**Line 88-92:** The DT products are reanalysis datasets that incorporate the highest quality altimeter measurements and geophysical corrections to minimize the risk of mass loss or false signals over time. The NRT data provide ready-to-use, real-time published altimeter data from all available missions. In the data processing, the DT products are computed optimally using a centered computation time window of $\pm 6$ weeks around the date of the map to be computed. In the NRT processing, future data are not available; therefore, the computation time window covers the period from 7 weeks prior to the computation date.

**Line 78-80:** The first ADT product (Fig. 1b) was produced using a ±11 days time window Near Real-Time (NRT) Two-Dimensional Variation (2DVAR) method with a 1/12° grid resolution (https://doi.org/10.5281/zenodo.11219285).

Eddy identification section:

Since most of the results depend on the eddy identification algorithm, it would be helpful to provide additional details about the methodology employed in your study. Specifically, could you elaborate on the method proposed by Ni (2014)? It seems challenging to find this reference－would you be able to provide a DOI or further citation details? Additionally, I would appreciate it if you could clarify the specific time period during which the eddy identification and tracking were conducted.

**Response:** Thank you for raising this question. It has indeed highlighted a basic mistake—we failed to correctly cite the primary source when referencing the eddy identification method. Upon re-examining the article by Ni (2014), we found that the eddy identification method used in that study was based on the work by Chelton et al. (2011). Therefore, we have cited the primary source, Chelton et al. (2011), in this context. Additionally, the time period for eddy identification and tracking has been added to Section 2.2.

**Line 116-119:** Therefore, this research employed a sea surface topography method based on contour analysis for eddy identification in 2DVAR and AVISO ADT merged maps, as well as in SWOT maps during both phases of the SWOT mission (Chelton et al., 2011). The eddy tracking was adopted only during science phase because of the fixed observation area of CALVAL phase.

Eddy validation section:

Since the eddy detection is specifically applied to the SWOT swath, it might be beneficial to undertake further validation to ensure its robustness. Conducting an intercomparison with independent datasets, such as SST or chlorophyll data, could provide valuable insights and help validate the eddy detection within the SWOT track. Additionally, the eddy detection performed along the SWOT track is inherently constrained by the SWOT swath extension, which is limited to 120 km. So, the eddy contours identified within the SWOT track are, by design, confined to this 120 km width. This raises an important question: what if the detected eddy contour corresponds to an isocontour of a larger eddy observed in broader maps (let's say the upper part of a large eddy)? In other words, how reliable is the eddy detection within the SWOT track? Is the SWOT eddy contour really a true eddy boundary? Complementary investigations and discussions on this aspect should be carried out to further validate the eddy detection in SWOT products.

**Response:** Thank you for your valuable suggestions. Indeed, using SWOT for eddy detection is an exploratory endeavor. Currently, there is no perfect method to verify the accuracy of SWOT eddies, and we have considered using independent datasets for validation. However, sea surface temperature data lacks the capability to identify eddy structures, and chlorophyll data is also merged map. Both sea surface temperature and chlorophyll data are remote sensing products, and their accuracy is not higher than that of SWOT. Using these types of data may introduce additional errors. On the other hand, considering that the SWOT swath data already have a good ability to identify fine-scale features (Fu et al., 2024; Martin et al., 2024; Ubelmann et al., 2024; Verger-Miralles et al., 2024; Zhang et al., 2024), we believe that our current method is sufficient for validating eddies in the merged maps.

Results section:

It would be nice to have a figure with the difference of ADT between SWOT and the AVISO & 2DVAR products to illustrate the amplitude difference between observed SWOT measurement and reconstructed NRT & DT AVISO and 2DVAR products.

**Response:** Thank you for your valuable suggestions. We have generated a figure that illustrates the distribution of differences between the sea surface height (SSH) from the merged fields and that from SWOT data (S1). The SSH values in both merged maps are consistently lower than those from SWOT by 0 – 10 cm and exhibit a degree of variability (seems like internal waves). Considering the coherence of the article and the value it brings, I would appreciate your help in deciding whether this figure merits inclusion in the main text.

[Figure]

S1 Differences in Absolute Dynamic Topography (ADT) between the 2DVAR and AVISO products compared with that of the SWOT data.

Since the results rely on eddy matching between the SWOT track and the altimetry gridded product, it might be helpful to include an illustration of eddy detection for both the SWOT track and the altimetry gridded products, as shown in Figure 3. Or superimpose contour of detected eddies between SWOT products and altimetry gridded products

**Response:** Thank you for your suggestion. We have incorporated the eddies identified by SWOT into Figure 3 in the main text, overlaying them with the eddies identified by the merged products (see S2).

[Figure]

S2 The ADT observation data of SWOT with the eddies detected by SWOT (in red), 2DVAR (in black, a1-c1) and AVISO (in black, a2-c2). The solid line represents the anticyclonic eddies and dashed line represents the cyclonic eddies.

Additionally, incorporating classical statistical metrics (in addition to eddy detection metrics) to evaluate the gridded product at a finer scale would enhance the analysis. For instance, presenting results based on RMSE comparisons between the gridded product and L3 SWOT could provide valuable insight on the error level in the products. This would also help to illustrate and intercompare the performance between the AVISO maps and the 2DVAR maps in the region of interest.

**Response:** Thank you for your valuable suggestions. We have generated a map showing the RMSE distribution between the two merged maps and SWOT (S3). The majority of the RMSE values fall within the range of 0.04–0.08 m, similar to the difference distribution, and still reveal internal wave signals. Considering the coherence of the manuscript and the value added by this figure, I would appreciate your help in deciding whether this figure merits inclusion in the main text.

[Figure]

S3 RMSE of ADT for 2DVAR and AVISO compared with SWOT data.

In the abstract you mentioned that: "The findings indicate that SWOT provides an enhanced capability in resolving fine-scale and mesoscale eddies in the South China Sea compared with conventional in-situ data, such as drifting buoys." Could you clarify which result supports this finding?

**Response:** Thank you very much for raising the issue. We indeed do not have sufficient data to support this conclusion; it was merely an example to illustrate that SWOT is more effective in detailed eddy boundary comparisons due to its larger data volume and extremely high accuracy. Therefore, we have revised the abstract.

**Line 24-25:** The SWOT data possess a greater potential for conducting detailed comparisons of eddy boundaries across fine-scale to mesoscale structures compared with conventional in-situ data, such as drifting buoys.

Discussion section:

To improve the Discussion section, it would be valuable to address several key points.

- Could you elaborate on the limitations of AVISO maps in terms of their resolution within the region of interest? Additionally, it would be helpful to discuss the differences between Near Real-Time (NRT) and Delayed Time (DT) products, as well as their implications for the 2DVAR approach.

**Response:** Thank you for raising this valuable question. In our prior research, we computed the effective resolution for both the AVISO and 2DVAR products within the study region. The effective resolution indicates the minimum spatial scale of signals that the merged maps can theoretically resolve, although it does not necessarily correspond to the actual minimum scale. The average effective resolution of AVISO in the SCS is approximately 150 km, whereas that of 2DVAR is about 80 km. This

suggests that, typically, AVISO identifies larger eddies compared to 2DVAR. As a result, AVISO encounters greater limitations in identifying eddies across the mesoscale to fine-scale spectrum.

We have included this discussion in the relevant section. The distinctions between the Near Real-Time (NRT) and Delayed Time (DT) products have been detailed in Section 2.1.

**Line 333-337:** The effective resolution indicates the minimum spatial scale of signals that the merged maps can theoretically resolve, although it does not necessarily correspond to the actual minimum scale. The average effective resolution of AVISO in the SCS is approximately 150 km, whereas that of 2DVAR is about 80 km. This suggests that, typically, AVISO identifies larger eddies compared to 2DVAR. As a result, AVISO encounters greater limitations in identifying eddies across the mesoscale to fine-scale spectrum.

**Line 88-92:** The DT products are reanalysis datasets that incorporate the highest quality altimeter measurements and geophysical corrections to minimize the risk of mass loss or false signals over time. The NRT data provide ready-to-use, real-time published altimeter data from all available missions. In the data processing, the DT products are computed optimally using a centered computation time window of ±6 weeks around the date of the map to be computed. In the NRT processing, future data are not available; therefore, the computation time window covers the period from 7 weeks prior to the computation date.

   - Could you explore potential reasons why only a limited number of eddies are detected in AVISO maps? This analysis would provide an important context for understanding the performance of different methodologies. Emphasizing the significance of developing regional configurations for merged sea surface height (SSH) products would also be a valuable addition, highlighting the potential for more accurate regional analyses.

**Response:** Your question is excellent, and we apologize for not addressing it in the text. The potential reason for the limited number of eddies detected in the AVISO maps is closely related to our previous work (Liu et al., 2023). Considering the small error correlation scales in high eddy kinetic energy regions such as the SCS and other coastal areas, the rational selection of the merged product's regional configuration is also very important. In terms of the trade-off between result performance, the background field time window for AVISO is selected as a multi-year average field, which leads to an increase in the scale of background error signals, and the scale of signals will be amplified during the mapping process. Therefore, the merged map has difficulty in reconstructing and identifying small-scale processes such as small-scale eddies, and will identify more large and mesoscale eddies. Due to the narrow swath of the SWOT track, the number of large and mesoscale eddies that can be identified by AVISO within this range is limited, which may ultimately lead to the limited number

of eddies detected by AVISO.

**Line 337-346:** Considering the small error correlation scales in high eddy kinetic energy regions such as the SCS and other coastal areas, the rational selection of the merged product's regional configuration is also very important. In terms of the trade-off between result performance, the background field time window for AVISO is selected as a multi-year average field, which leads to an increase in the scale of background error signals, and the scale of signals will be amplified during the mapping process. Therefore, the merged map has difficulty in reconstructing and identifying small-scale processes such as small-scale eddies, and will identify more large and mesoscale eddies. Due to the narrow swath of the SWOT track, the number of large and mesoscale eddies that can be identified by AVISO within this range is limited, which may ultimately lead to the limited number of eddies detected by AVISO. In contrast, 2DVAR adopts a one-day background field time window, and the reduction in background error correlation scales allows for the reconstruction of more fine-scale to mesoscale signals.

- It would also be useful to consider the broader implications of your study. Are the conclusions drawn from your research applicable on a global scale? Could the proposed method be effectively applied at a global level? Do the findings regarding AVISO maps and the 2DVAR approach remain consistent at a global scale? Addressing these questions would provide a more comprehensive perspective. I would suggest for example performing similar analysis at global scale using CMEMS NRT & DT product and L3 SWOT data to illustrate the limitation of NRT products and test the method at global scale.

**Response:** Thank you for raising the question. Currently, work on a global scale is still in progress. Therefore, we can only limit the scope of this study to the South China Sea for now and revise the title to "Advances in Surface Water and Ocean Topography for Fine-Scale Eddy Identification from Altimeter Sea Surface Height Merging Maps in the South China Sea."

- In Figure 6, the eddies appear to remain relatively static over the 2–3 months of tracking analysis. It would be helpful if the authors could discuss whether these eddy structures might also be detectable or sampled by classical nadir altimeters and elaborate on the added value that SWOT brings compared to traditional nadir altimeters. What are (on average) observable length scale of nadir altimeter in this region? Maybe an illustration of all nadir track over specific region may highlight the difference or similarities between swot 2d product and nadir 1d products.

**Response:** Thank you very much for your valuable suggestions. Our previous description indeed failed to clearly highlight the differences between SWOT and traditional nadir altimeters. In fact, traditional nadir altimeters, due to their linear single-point sampling method, lack the capability to identify ocean eddies, which

have a two-dimensional structure. To illustrate the distinction between SWOT's KaRIn and nadir observations, we have added the original nadir data in Fig. 1a and provided a zoomed-in view to clearly show the sampling intervals and effects of different observation methods. We added relative discussion in the relevant section about the differences in eddy structure detection between SWOT and traditional altimeters and the added value that SWOT can provide.

**Line 365-368:** Traditional nadir altimeters, due to their linear single-point sampling method, lack the capability to identify ocean eddies, which have a two-dimensional structure. Therefore, compared to traditional nadir altimeters, the advantage of SWOT lies in its ability to quantitatively analyze the observable scales of two-dimensional structures, such as eddies, within the study area.

    - Including a section that describes the importance of the South China Sea (SCS) could also further strengthen the motivation for this study

**Response:** Thank you for your valuable suggestions. The significance of the South China Sea (SCS) as the study area has been added to the last paragraph of Section 2.1.

**Line 98-106:** The SCS is a significant dynamic marginal sea in the northwestern Pacific, featuring complex bathymetry, a large area, and multiple straits that facilitate water exchange with the Pacific and Indian Oceans(Chen et al., 2023). It serves as an exemplary model of an open ocean with well-defined continental shelves, shelf breaks, and a central deep basin. In the SCS, the first obliquely pressured Rossby deformation radius was less than 20 km in winter (Cai et al., 2008), suggesting a rich environment for fine-scale oceanic dynamical processes. Additionally, the SCS receives energy transport from the sub-mesoscale energy reservoir of the Kuroshio Current via the western boundary currents, resulting in a dense concentration of mesoscale and fine-scale processes on the 10-km scale (Lin et al., 2020; Ni et al., 2021; Zu et al., 2019). Thus, this study of the SCS holds substantial significance and reference value for understanding complex dynamic marginal seas and the broader northwestern Pacific region.

---

## Referee Report (RR1)

Review of manuscript egusphere-2024-2773 entitled " Advances in Surface Water and Ocean Topography for Fine-scale Eddy Identification from Altimeter Sea Surface Height Merging Maps in the South China Sea"

**Main comment:**
**I thoroughfully read the author responses to both reviewer and I think they globally addressed the issues in a satisfactory way. In my opinion the paper has improved and is a good contribution for the community to assess the performances of the 2DVAR method. I therefore recommend publication after minor comments have been addressed.**

1) Thanks for using a later version of SWOT data. I noticed that some figures have changed from previous version (Fig. 3 and 4) by selecting other dates as examples. Is this due to the difference in SWOT version ?

2) Section 3.2 still lack from statistical evidences. As far as I am concerned, it only shows few visual examples of section 3.1 results. For instance, the dates of the data shown are completely arbitrary (or not ?) so one might wonder how would it look like if another period is taken. Also only one drifter trajectory is used so one might think that the authors only show an example of 2DVAR performing better than AVISO. Can we find examples of AVISO performing better than 2DVAR ? Presented as is, it seems the authors chose the right example to illustrate what they want to demonstrate but is it statistically true ? I do not question the authors scientific integrity here but it seems to me this section provides insufficient evidences as compared to the other sections presenting more robust (statistically) results.

L 24-25: "The SWOT data provide a greater potential for resolving fine-scale to mesoscale eddies in the South China Sea compared with conventional in-situ data, such as drifting buoys." This sentence has not been changed in the revised manuscript although it is in the "diff" file. The response given to Reviewer#2 is clear and the suggested modification should appear in the revised version of the manuscript.

L 59-60: "Fine-scale..": I think this sentence would better fit in the text at lines 32-33.

L63: "diverse merging methods". Please specify here which one will be tested in the study for clarity.

L80: "11219285" Is it a reference ?

L84: Specifically they include SWOT nadir data.

Figure 1: Caption should be updated since on the new version (a) SWOT, (b) 2DVAR …

L126: Typo: "a difference".

Figure 3 and 4: How did the author fill the gap between SWOT swaths (linear interpolation..)? Also in the caption of Figure 4 it is mentioned that nadir data from swot are used, however on lines 75-77 it is stated they were excluded.

L280: "However, due to being constantly entrained by a single eddy and rotating within it, the number of eddies that can be detected by a drifter buoy is limited compared to the rapid mapping provided by SWOT". These observation platforms are inherently different so by definition a single buoy cannot sample as many eddies as SWOT. However, by using a sufficiently high number of drifters one can catch a similar number of eddies as SWOT. This sentence is not relevant, I suggest to remove it.
Figure 7: How did the authors get a gridded SWOT map of eddy radius? Is using several SWOT tracks filling the entire area ? The methodology here needs to be clearly stated.

L322: "the method is validated through in-situ observations". If the authors refers to SWOT observations I would not use "in situ". If the authors refer to the drifter observation, it is only one observation.

L353: Typo: "ot"

---

## Author Response (AR2)

**Reply on Report #1**

*Thank you for your new comments concerning our manuscript. Those comments are all valuable and very helpful for revising and improving our paper. We have studied comments carefully and have made correction which we hope meet with your approval. The main corrections in the paper and the responses to your comments are highlighted in blue and are as follows:*

1) Thanks for using a later version of SWOT data. I noticed that some figures have changed from previous version (Fig. 3 and 4) by selecting other dates as examples. Is this due to the difference in SWOT version ?

**Response:** Thank you for your question, and we apologize for any lack of clarity in our previous explanation. Regarding Figure 3, we recognize that the initially selected dates (September 12, September 13, and September 17) were too close together, which did not fully utilize the entire SWOT dataset. In addition, although the height map in Figure 3c (September 17) of the previous version showed some eddy-like features, the eddy detection algorithm was not able to successfully capture eddies consistent with the merged map due to limitations in the SWOT data. To ensure a broader representation of satellite cycles and spatial variability than the last version, we have increased the temporal intervals between the selected cases. Therefore, we have reselected examples from September 7, September 12, and September 30 for demonstration and have revised the corresponding results description, which is provided below.

As for Figure 4, following Reviewer #2's suggestion, we added the SWOT-derived eddies to the figure and found that shifting the display by one day better captures the temporal evolution of both cyclonic and anticyclonic eddies. The previous version did not sufficiently illustrate the development of the cyclonic eddy.

**Line 245-254:** The 2DVAR product demonstrates strong agreement with SWOT-derived anticyclonic eddies at 116°E, 21°N in Fig. 3(a1), 112°E, 17.5°N and 112°E, 16°N in Fig. 3(b1), and 113°E, 18°N in Fig. 3(c1), as well as with the cyclonic eddy at 111.5°E, 16.5°N in Fig. 3(b1). In contrast, the AVISO product fails to accurately match the cyclonic eddy at 111.5°E, 16.5°N with the SWOT observations in Fig. 3(b2). Additionally, the AVISO product captures an eddy at 116°E, 21°N in Fig. 3(a2) that only partially overlaps with the corresponding SWOT eddy, with minimal spatial correspondence. For other eddies where both merged products exhibit agreement, the radii of the AVISO eddies are notably larger than those identified by 2DVAR. Although the AVISO product exhibits lower consistency with SWOT observations compared to 2DVAR, it still captures a considerable number of eddies that align with SWOT, thereby maintaining its fundamental utility as a merged product for eddy identification. However, both products fail to detect certain small eddies in SWOT observations, such as the cyclonic eddy at 112.5°E, 17°N in

Fig. 3(c).

**Line 270-279:** In the 2DVAR maps, two closely spaced mesoscale anticyclones were identified at 110°E, 15°N, and 110.5°E, 16.2°N in Figs. 4(a1) – (e1), along with an anticyclone at 110.5°E, 13.2°N in Figs. 4(b1, d1 and e1), and a smaller-scale cyclone at 110.5°E, 14.8°N in Figs. 4(b1) – (e1). These eddies derived from 2DVAR exhibit discrepancies when compared to the SWOT-derived eddies, particularly in the case of the eddy at 110.5°E, 16.2°N whose radius varies daily in the 2DVAR results and represented by a colored circle on the map rather than being identified as a distinct eddy in the SWOT data. Similar to as Fig. 3, the 2DVAR method outperforms AVISO, which erroneously merges the two closely spaced anticyclones into a single larger eddy and fails to capture the eddies at 110.5°E, 14.8°N and 110.5°E, 13.2°N. Notably, by accurately matching the emergence and dissipation of SWOT eddies at 110.5°E, 14.8°N and 110.5°E, 13.2°N, the 2DVAR method demonstrates its capability to reconstruct eddies that evolve over time, despite some relative positional deviations from the actual eddy location.

2) Section 3.2 still lack from statistical evidences. As far as I am concerned, it only shows few visual examples of section 3.1 results. For instance, the dates of the data shown are completely arbitrary (or not ?) so one might wonder how would it look like if another period is taken. Also only one drifter trajectory is used so one might think that the authors only show an example of 2DVAR performing better than AVISO. Can we find examples of AVISO performing better than 2DVAR ? Presented as is, it seems the authors chose the right example to illustrate what they want to demonstrate but is it statistically true ? I do not question the authors scientific integrity here but it seems to me this section provides insufficient evidences as compared to the other sections presenting more robust (statistically) results.

**Response:** It is certain that, in selecting the cases for Figures 3 and 4, we chose time points from the dataset that are representative of the eddy boundary reconstruction capabilities of both AVISO and 2DVAR. Although in these examples, AVISO performs less effectively than 2DVAR in representing eddy boundaries, it still retains the ability to capture eddies. The differences between AVISO and 2DVAR primarily lie in the accuracy of eddy radius and positional representation, which aligns with the statistical characteristics discussed in section 3.1. These examples represent the best cases we could select to illustrate these differences. The observed discrepancies do not arise from biased selection but rather reflect statistical differences in the eddy reconstruction performance between the two products, as demonstrated in Sections 3.1 and 3.3. Sections 3.1 and 3.3 comprehensively illustrate how AVISO eddies are statistically larger in scale compared to those identified by 2DVAR, while Section 3.2 focuses on a detailed comparison of the actual boundaries of individual eddies. The purpose of Figure 5 is primarily to highlight the differences in eddy assessment efficiency between drifter buoys and SWOT observations.

Similar to Figures 3 and 4, we aimed to select cases that are representative of both products. However, due to the limited spatial distribution and low observational frequency of the drifter data, the number of valid eddy detection cases available for analysis is highly constrained. In other instances, the differences between 2DVAR and AVISO in comparison with the drifter data are not significant (either both match the drifter trajectories or neither does). The case involving drifter 300534064134530 stands out as the only example demonstrating a clear and favorable comparison.

**Line 255-260:** It is important to emphasize that the examples presented in this section are representative of the eddy boundary reconstruction capabilities of both AVISO and 2DVAR. While these examples show that AVISO performs slightly worse than 2DVAR in terms of eddy radius and positional accuracy, they nonetheless represent some of the best-case scenarios for the AVISO product. This conclusion is not influenced by selective bias in the examples chosen but rather reflects the inherent performance differences between the two merged products, which is consistent with the statistical findings presented in section 3.1.

**Line 293-297:** Similar to Figures 3 and 4, we have endeavored to select cases of eddy detection that are representative for both products. However, due to the limited spatial distribution and low observational frequency of the drifter data, the number of valid eddy detection cases available for analysis is highly constrained. In other instances, the differences between 2DVAR and AVISO in comparison with the drifter data are not significant (either both match the drifter trajectories or neither does). The case involving drifter 300534064134530 stands out as the only example demonstrating a clear and favorable comparison.

3) L 24-25: "The SWOT data provide a greater potential for resolving fine-scale to mesoscale eddies in the South China Sea compared with conventional in-situ data, such as drifting buoys." This sentence has not been changed in the revised manuscript although it is in the "diff" file. The response given to Reviewer#2 is clear and the suggested modification should appear in the revised version of the manuscript.

**Response:** Thank you for your question. We acknowledge this oversight and have made the necessary revisions, including adjustments to the wording for improved clarity and precision.

**Line 24-25:** SWOT data are more likely to provide detailed comparisons of eddy boundaries for fine- to meso-scale structures compared with conventional in-situ data (e.g., drifting buoys).

4) L 59-60: "Fine-scale..": I think this sentence would better fit in the text at lines 32-33.

**Response:** Thank you for your valuable suggestions. We have incorporated the necessary revisions accordingly.

**Line32-33:** Fine-scale ocean processes are characterized by spatial variability of 1 to 100 kilometres and temporal variability of days to months (Lévy et al., 2024).

5) L63: "diverse merging methods". Please specify here which one will be tested in the study for clarity.

**Response:** Thank you for your valuable suggestions. We have incorporated the necessary revisions accordingly.

**Line 63-67:** This study aimed to validate the accuracy and reliability of different merging methods, specifically 2DVAR and AVISO, in reconstructing oceanic dynamic signals, with a particular focus on fine-scale eddies.

6) L80: "11219285" Is it a reference ?

**Response:** We apologize for the inclusion of redundant numbers, which has now been removed.

**Line 83-83:** The first ADT product (Fig. 1b) was produced using a $\pm 11$ days time window Near Real-Time (NRT) Two-Dimensional Variation (2DVAR) method with a $1/12°$ grid resolution.

7) L84: Specifically they include SWOT nadir data.

**Response:** Thank you for your correction. We have made the necessary revisions accordingly.

**Line 88-91:** During the science phase of the SWOT mission, the AVISO merged map Delayed Time (DT) products utilized SWOT nadir data as an input source (Copernicus Marine Service repository, 2023b). To maintain the independence of the datasets, we employed NRT products, which do not include SWOT nadir data (Copernicus Marine Service repository, 2023a).

8) Figure 1: Caption should be updated since on the new version (a) SWOT, (b) 2DVAR …

**Response:** Thank you for your correction. We have made the necessary revisions accordingly.

**Line 113-114:** Figure 1. Four datasets of absolute dynamic topography (ADT) in the

South China Sea. (a) SWOT, (b) 2DVAR, (c) AVISO, and (d) GDP. The ADT data in (a), (b), and (c) were obtained on September 12, 2023, and (d) covers the entire period of the Science phase.

9) L126: Typo: "a difference".

**Response:** Thank you for your correction. We have made the necessary revisions accordingly.

**Line 131-132:** All possible eddies with a difference in ADT between the eddy center and the boundary contour line of less than $\pm 2$ cm were excluded from further analysis.

10) Figure 3 and 4: How did the author fill the gap between SWOT swaths (linear interpolation..)? Also in the caption of Figure 4 it is mentioned that nadir data from swot are used, however on lines 75-77 it is stated they were excluded.

**Response:** Thank you for your question. We acknowledge this oversight and have made the necessary revisions. To ensure consistency in data resolution and to focus the current study on the fine-scale to mesoscale, we employed a regional averaging method to reduce the resolution of the SWOT data from the original 2-km sampling interval to 1/12°. Owing to the inclination angle between the SWOT satellite orbital plane and the equatorial plane, each downsampled square region is covered by observations. Consequently, interpolation across swath gaps is unnecessary, thereby avoiding the substantial errors that associated with interpolating these gaps.

We apologize for the oversight in the title of Figure 4, which has now been corrected.

**Line 78-82:** To ensure consistency in data resolution and to focus the current study on the fine-scale to mesoscale, we employed a regional averaging method to reduce the resolution of the SWOT data from the original 2-km sampling interval to 1/12°. Owing to the inclination angle between the SWOT satellite orbital plane and the equatorial plane, each downsampled square region is covered by observations. Consequently, interpolation across swath gaps is unnecessary, thereby avoiding the substantial errors that associated with interpolating these gaps.

**Line 265-267:** Figure 4. Observation data of SWOT (in red) with the 2DVAR (in black, a1-e1) and AVISO (in black, a2-e2) merged maps from 2023/04/06 to 04/10. The solid line (dashed) represents the anticyclonic (cyclonic) eddy, and the colour-filled plot contains the KaRIn data from SWOT.

11) L280: "However, due to being constantly entrained by a single eddy and rotating within it, the number of eddies that can be detected by a drifter buoy is limited compared to the rapid mapping provided by SWOT". These observation

platforms are inherently different so by definition a single buoy cannot sample as many eddies as SWOT. However, by using a sufficiently high number of drifters one can catch a similar number of eddies as SWOT. This sentence is not relevant, I suggest to remove it.

**Response:** Thank you for your suggestion. This sentence was indeed redundant and has now been removed.

12) Figure 7: How did the authors get a gridded SWOT map of eddy radius? Is using several SWOT tracks filling the entire area ? The methodology here needs to be clearly stated.

**Response:** Yes, we utilized data spanning several times the satellite observation period within the scientific phase, ensuring comprehensive coverage of the South China Sea region. Based on this dataset, we generated a distribution map of eddy radii.

**Line 305-307:** In this section, we utilized data spanning several times the satellite observation period within the scientific phase, ensuring comprehensive coverage of the South China Sea region. Based on this dataset, we generated a distribution map of eddy characteristics.

13) L322: "the method is validated through in-situ observations". If the authors refers to SWOT observations I would not use "in situ". If the authors refer to the drifter observation, it is only one observation.

**Response:** Thank you for your valuable feedback. The relevant content has now been removed.

**Line 340-341:** Finally, the method is validated through observations, ensuring robustness and reliability.

14) L353: Typo: "ot"

**Response:** Thank you for your correction. We have made the necessary revisions accordingly.

**Line 371-372:** Also, ignoration of non-closure of contour lines in SWOT maps might be a deficiency too.